# Estimating Transfer Entropy in Continuous Time Between Neural Spike Trains or Other Event-Based Data

**David P. Shorten**[1]*, **Richard E. Spinney**[1,2], **Joseph T. Lizier**[1]

**1** Complex Systems Research Group and Centre for Complex Systems, Faculty of Engineering, The University of Sydney, Sydney, Australia, **2** School of Physics and EMBL Australia Node Single Molecule Science, School of Medical Sciences, The University of New South Wales, Sydney, Australia

* david.shorten@sydney.edu.au

**Data Availability Statement:** The code for generating the datasets is available in a public repository: https://github.com/dpshorten/CoTETE_experiments.

## Abstract

Transfer entropy (TE) is a widely used measure of directed information flows in a number of domains including neuroscience. Many real-world time series for which we are interested in information flows come in the form of (near) instantaneous events occurring over time. Examples include the spiking of biological neurons, trades on stock markets and posts to social media, amongst myriad other systems involving events in continuous time throughout the natural and social sciences. However, there exist severe limitations to the current approach to TE estimation on such event-based data via discretising the time series into time bins: it is not consistent, has high bias, converges slowly and cannot simultaneously capture relationships that occur with very fine time precision as well as those that occur over long time intervals. Building on recent work which derived a theoretical framework for TE in continuous time, we present an estimation framework for TE on event-based data and develop a *k*-nearest-neighbours estimator within this framework. This estimator is provably consistent, has favourable bias properties and converges orders of magnitude more quickly than the current state-of-the-art in discrete-time estimation on synthetic examples. We demonstrate failures of the traditionally-used source-time-shift method for null surrogate generation. In order to overcome these failures, we develop a local permutation scheme for generating surrogate time series conforming to the appropriate null hypothesis in order to test for the statistical significance of the TE and, as such, test for the conditional independence between the history of one point process and the updates of another. Our approach is shown to be capable of correctly rejecting or accepting the null hypothesis of conditional independence even in the presence of strong pairwise time-directed correlations. This capacity to accurately test for conditional independence is further demonstrated on models of a spiking neural circuit inspired by the pyloric circuit of the crustacean stomatogastric ganglion, succeeding where previous related estimators have failed.

**Funding:** JL was supported through the Australian Research Council DECRA grant DE160100630 - https://www.arc.gov.au/grants/discovery-program/discovery-early-career-researcher-award-decra and The University of Sydney Research Accelerator (SOAR) Fellowship program - https://sydney.edu.au/research/our-researchers/sydney-research-accelerator-fellows.html. The funders had no role in study design, data collection and analysis, decision to publish, or preparation of the manuscript.

**Competing interests:** The authors have declared that no competing interests exist.

## Author summary

Transfer Entropy (TE) is an information-theoretic measure commonly used in neuroscience to measure the directed statistical dependence between a source and a target time series, possibly also conditioned on other processes. Along with measuring information flows, it is used for the inference of directed functional and effective networks from time series data. The currently-used technique for estimating TE on neural spike trains first time-discretises the data and then applies a straightforward plug-in information-theoretic estimation procedure. This approach has numerous drawbacks: it has high bias, cannot capture relationships occurring on both fine and large timescales simultaneously, converges very slowly as more data is obtained, and indeed does not even converge to the correct value for any practical non-vanishing discretisation scale. We present a new estimator for TE which operates in continuous time and demonstrate, via application to synthetic examples, that it addresses these problems and can reliably differentiate statistically significant flows from (conditionally) independent spike trains. Further, we also apply it to more biologically-realistic spike trains obtained from a biophysical model inspired by the pyloric circuit of the crustacean stomatogastric ganglion; our correct inference of directed conditional dependence and independence between neurons here provides an important validation for our approach where similar methods have previously failed.

This is a *PLOS Computational Biology* Methods paper.

## Introduction

In analysing time series data from complex dynamical systems, such as in neuroscience, it is often useful to have a notion of information flow. We intuitively describe the activities of brains in terms of such information flows: for instance, information from the visual world must flow to the visual cortex where it will be encoded [1]. Further, information coded in the motor cortex must flow to muscles where it will be enacted [2].

Transfer entropy (TE) [3, 4] has become a widely accepted measure of such flows. It is defined as the mutual information between the past of a source time-series process and the present state of a target process, conditioned on the past of the target. More specifically (in discrete time), the transfer entropy rate [5] is:

$$\dot{\mathbf{T}}_{Y \to X} = \frac{1}{\Delta t} I(X_t ; \mathbf{Y}_{<t} \mid \mathbf{X}_{<t}) = \frac{1}{\tau} \sum_{t=1}^{N_T} \ln \frac{p(x_t \mid \mathbf{x}_{<t}, \mathbf{y}_{<t})}{p(x_t \mid \mathbf{x}_{<t})}. \tag{1}$$

Here the information flow is being measured from a source process $Y$ to a target $X$, $I(\cdot;\cdot|\cdot)$ is the conditional mutual information [6], $p(\cdot|\cdot)$ is a conditional probability, $x_t$ is the current state of the target, $\mathbf{x}_{<t}$ is the history of the target, $\mathbf{y}_{<t}$ is the history of the source, $\Delta t$ is the interval between time samples (in units of time), $\tau$ is the length of the time series and $N_T = \tau/\Delta t$ is the number of time samples. The histories $\mathbf{x}_{<t}$ and $\mathbf{y}_{<t}$ are usually captured via embedding vectors, e.g. $\mathbf{x}_{<t} = \mathbf{x}_{t-m:t-1} = \{x_{t-m}, x_{t-m+1}, \ldots, x_{t-1}\}$. The average here is taken over time, as opposed to possible states and histories (both formulations are equivalent under the assumptions of stationarity and ergodicity). Recent work [5] has highlighted the importance of normalising the TE by the width of the time bins, as above, such that it becomes a *rate*, in order to ensure convergence in the limit of small time bin size.

It is also possible to condition the TE on additional processes [4]. Given additional processes $\boldsymbol{\mathcal{Z}} = \{Z_1, Z_2, \ldots, Z_{n_{\boldsymbol{\mathcal{Z}}}}\}$ with histories $\boldsymbol{\mathcal{Z}}_{<t} = \{\mathbf{Z}_{1,<t}, \mathbf{Z}_{2,<t}, \ldots, \mathbf{Z}_{n_{\boldsymbol{\mathcal{Z}}},<t}\}$, we can write the conditional TE rate as

$$\dot{\mathbf{T}}_{Y \to X | \boldsymbol{\mathcal{Z}}} = \frac{1}{\Delta t} I(X_t \,;\, \mathbf{Y}_{<t} \,|\, \mathbf{X}_{<t}, \boldsymbol{\mathcal{Z}}_{<t}). \tag{2}$$

When combined with a suitable statistical significance test, the TE (and conditional TE) can be used to show that the present state of $X$ is conditionally independent of the past of $Y$– when conditioned on the past of $X$ (and on the conditional processes $\boldsymbol{\mathcal{Z}}$). Of course, we refer to conditional independence in the statistical sense (i.e. $p(x_t \,|\, \mathbf{x}_{<t}, \boldsymbol{\varkappa}_{<t}, \mathbf{y}_{<t}) = p(x_t \,|\, \mathbf{x}_{<t}, \boldsymbol{\varkappa}_{<t})$) rather than the causal sense. Such a conditional independence test can be used as a component in a network inference algorithm and, as such, TE is widely used for inferring directed functional and effective network models [7, 8, 9, 10, 11, 12] (and see [4, Sec. 7.2] for a review).

TE has enjoyed widespread application in neuroscience in particular [13, 14]. Uses have included the functional/effective network inference as mentioned above, as well as the measurement of the direction and magnitude of information flows [15, 16] and the determination of transmission delays [17]. Such applications have been performed using data from multiple diverse sources such as MEG [18, 19], EEG [20], fMRI [21], electrode arrays [22], calcium imaging [9] and simulations [23].

Previous applications of TE to spike trains [22, 24, 25, 26, 27, 28, 29, 30, 31, 32, 33, 34] and other types of event-based data [35], including for the purpose of network inference [9, 36, 37], have relied on time discretisation. As shown in Fig 1, the time series is divided into small bins of width $\Delta t$. The value of a sample for each bin could then be assigned a binary value—denoting the presence or absence of events (spikes) in the bin—or a natural number denoting the number of events (spikes) that fell within the bin (the experiments in this paper use the former). A choice is made as to the number of time bins, $l$ and $m$, to include in the source and target history embeddings $\mathbf{y}_{<t}$ and $\mathbf{x}_{<t}$. This results in a finite number of possible history embeddings. For a given combination $\mathbf{x}_{<t}$ and $\mathbf{y}_{<t}$, the probability of the target's value in the current bin conditioned on these histories, $p(x_t | \mathbf{x}_{<t}, \mathbf{y}_{<t})$, can be directly estimated using the plugin (histogram) [38] estimator. The probability of the target's value in the current bin conditioned on only the target history, $p(x_t | \mathbf{x}_{<t})$, can be estimated in the same fashion. From these estimates the TE can be calculated in a straightforward manner via Eq (1). See Results for a description of the application of the discrete time TE estimator to synthetic examples including spiking events from simulations of model neurons.

There are two large disadvantages to this approach [5]. If the process is genuinely occurring in discrete time, then the estimation procedure just described is consistent. That is, it is guaranteed to converge to the true value of the TE in the limit of infinite data. However, if we are considering a fundamentally continuous-time process (with full measurement precision), such as a neuron's action potential, then the lossy transformation of time discretisation ($\Delta t > 0$) will result in an inaccurate estimate of the TE. Thus, in these cases, any estimator based on time discretisation is not consistent. Secondly, whilst the loss of resolution of the discretization will reduce with decreasing bin size $\Delta t$, this requires larger dimensionality in the history embeddings to capture correlations over similar time intervals. This increase in dimension will result in an exponential increase in the state space size being sampled to estimate $p(x_t | \mathbf{x}_{<t}, \mathbf{y}_{<t})$, and therefore the data requirements. However, some recordings of the activities of neurons are done with low time precision. For example, recordings from calcium imaging experiments usually use a sampling rate of around 1 to 10 Hz [39]. In such cases, we could use bin sizes on the order of the experimental precision and still capture a reasonable history

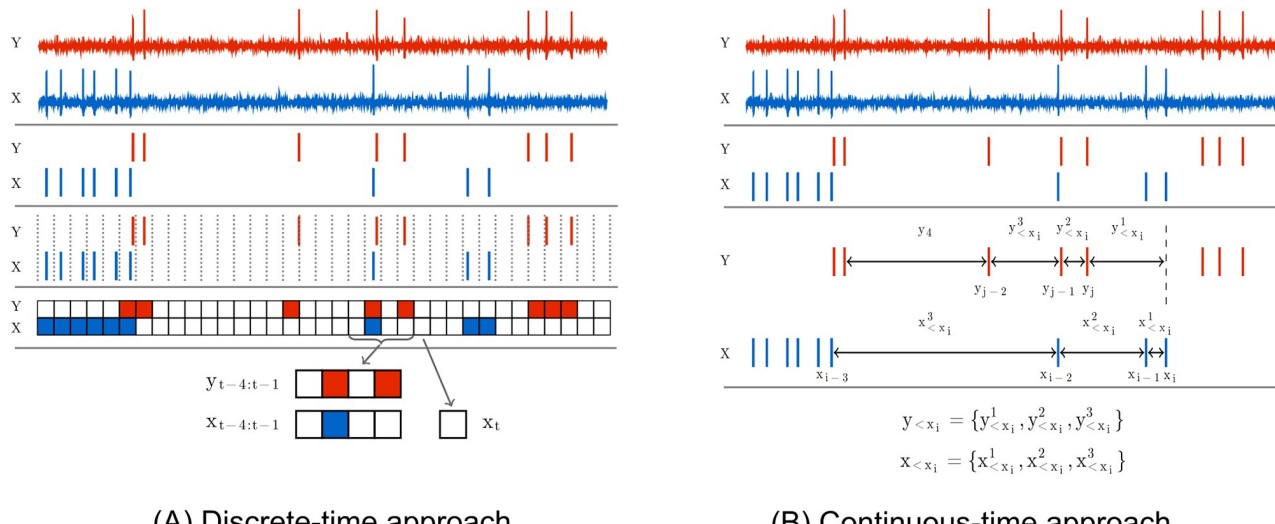

**Fig 1. Diagrams highlighting the differences in the embeddings used by the discrete and continuous-time estimators.** The discrete-time estimator (A) divides the time series into time bins. A binary value is assigned to each bin denoting the presence or absence of a (spiking) event–alternatively, this could be a natural number to represent the occurrence of multiple events. The process is thus recast as a sequence of binary values and the history embeddings ($\mathbf{x}_{t-4:t-1}$ and $\mathbf{y}_{t-4:t-1}$) for each point are binary vectors. The probability of an event occurring in a bin, conditioned on its associated history embeddings, is estimated via the plugin (histogram) [38] estimator. Conversely, the continuous-time estimator (B) performs no time binning. History embeddings $\mathbf{x}_{<x_i}$ and $\mathbf{y}_{<x_i}$ for events or $\mathbf{x}_{<u_i}$ and $\mathbf{y}_{<u_i}$ for arbitrary points in time (not shown in this figure, see Fig 10) are constructed from the raw interspike intervals. This approach estimates the TE by comparing the probabilities of the history embeddings of the target processes' history as well as the joint history of the target and source processes at both the (spiking) events and arbitrary points in time.

length with history embeddings composed of only a small number of bins. This might keep the size of the history state space small enough that we can collect an adequate number of samples for each history permutation with the available data. In such cases, we might expect the discrete-time approach to perform as well as can be expected given the limitations imposed by the apparatus. On the other hand, data from microelectrode arrays can be sampled at rates over 70 kHz [40]. When using data collected with this high temporal precision, if we use bin sizes corresponding to the sampling rate, we will be forced to use incredibly short history embeddings in order to avoid the size of the history state space growing to a point where it can no longer be sampled.

In practice then, if the data has been collected with fine temporal precision, the application of transfer entropy to event-based data such as spike trains has often required a trade-off between fully resolving interactions that occur with fine time precision and capturing correlations that occur across long time intervals. There is substantial evidence that spike correlations at the millisecond and sub-millisecond scale play a role in encoding visual stimuli [41, 42], motor control [43] and speech [44]. On the other hand, correlations in spike trains exist over lengths of hundreds of milliseconds [45]. A discrete-time TE estimator cannot capture both of these classes of effects simultaneously, and remains heavily dependent on the value of $\Delta t$ [31].

Recent work by Spinney et al. [5] derived a continuous-time formalism for TE. It was demonstrated that, for stationary point processes such as spike trains, the pairwise TE rate is given by:

$$\dot{\mathbf{T}}_{Y \to X} = \lim_{\tau \to \infty} \frac{1}{\tau} \sum_{i=1}^{N_X} \ln \frac{\lambda_{x|\mathbf{x}_{<t}, \mathbf{y}_{<t}}[\mathbf{x}_{<x_i}, \mathbf{y}_{<x_i}]}{\lambda_{x|\mathbf{x}_{<t}}[\mathbf{x}_{<x_i}]}. \tag{3}$$

Here $N_X$ is the number of events in the target process and $\tau$ is the length in time of this process

whilst $\lambda_{x|\mathbf{x}_{<t}, \mathbf{y}_{<t}}[\mathbf{x}_{<x_i}, \mathbf{y}_{<x_i}]$ is the instantaneous firing rate of the target conditioned on the histories of the target $\mathbf{x}_{<x_i}$ and source $\mathbf{y}_{<x_i}$ at the time points $x_i$ of the events in the target process. $\lambda_{x|\mathbf{x}_{<t}}[\mathbf{x}_{<x_i}]$ is the instantaneous firing rate of the target conditioned on its history alone, ignoring the history of the source. Note that $\lambda_{x|\mathbf{x}_{<t}, \mathbf{y}_{<t}}$ and $\lambda_{x|\mathbf{x}_{<t}}$ are defined at all points in time and not only at target events. It is worth emphasizing that, in this context, the processes $X$ and $Y$ are series of the time points $x_i$ and $y_j$ of the events $i$ and $j$ in the target and source respectively. This is contrasted with Eq (1), where $X$ and $Y$ are time series of values at the sampled time points $t_i$. To avoid confusion we use the notation that the $y_j \in Y$ are the raw time points and $\mathbf{y}_{<x_i}$ is some representation of the history of $Y$ observed at the time point $x_i$ (see Methods).

Eq (3) can easily be adapted to the conditional case:

$$\dot{\mathbf{T}}_{Y \rightarrow X | \boldsymbol{\mathcal{Z}}} = \lim_{\tau \rightarrow \infty} \frac{1}{\tau} \sum_{i=1}^{N_X} \ln \frac{\lambda_{x|\mathbf{x}_{<t}, \mathbf{y}_{<t}, \boldsymbol{\varkappa}_{<t}}[\mathbf{x}_{<x_i}, \mathbf{y}_{<x_i}, \boldsymbol{\varkappa}_{<x_i}]}{\lambda_{x|\mathbf{x}_{<t}, \boldsymbol{\varkappa}_{<t}}[\mathbf{x}_{<x_i}, \boldsymbol{\varkappa}_{<x_i}]}. \tag{4}$$

Here $\lambda_{x|\mathbf{x}_{<t}, \mathbf{y}_{<t}, \boldsymbol{\varkappa}_{<t}}[\mathbf{x}_{<x_i}, \mathbf{y}_{<x_i}, \boldsymbol{\varkappa}_{<x_i}]$ is the instantaneous firing rate of the target conditioned on the histories of the target $\mathbf{x}_{<x_i}$, source $\mathbf{y}_{<x_i}$ and other possible conditioning processes $\boldsymbol{\varkappa}_{<x_i} = \{\mathbf{z}_{1,<x_i}, \mathbf{z}_{2,<x_i}, \ldots, \mathbf{z}_{n_{\varkappa}, <x_i}\}$. $\lambda_{x|\mathbf{x}_{<t}, \boldsymbol{\varkappa}_{<t}}[\mathbf{x}_{<x_i}, \boldsymbol{\varkappa}_{<x_i}]$ is the instantaneous firing rate of the of the target conditioned on the histories of the target and the additional conditioning processes, ignoring the history of the source.

Crucially, it was demonstrated by Spinney et al., and later shown more rigorously by Cooper and Edgar [46], that if the discrete-time formalism of the TE (in Eq (1)) could be properly estimated as $\lim_{\Delta t \rightarrow 0}$, then it would converge to the same value as the continuous-time formalism. This is due to the contributions to the TE from the times between target events vanishing in expectation. Yet there are two important distinctions in the continuous-time formalism which hold promise to address the consistency issues of the discrete-time formalism. Firstly, the basis in continuous time allows us to efficiently represent the history embeddings by inter-event intervals, suggesting the possibility of jointly capturing subtleties in both short and long time-scale effects that has evaded discrete-time approaches. See Fig 1 for a diagrammatic representation of these history embeddings, contrasted with the traditional way of constructing histories for the discrete-time estimator. Secondly, note the important distinction that the sums in Eqs (3) and (4) are taken over the $N_X$ (spiking) events in the target during the time-series over interval $\tau$; this contrasts to a sum over all time-steps in the discrete-time formalism. An estimation strategy based on Eqs (3) and (4) would only be required to calculate quantities at events, ignoring the inter-event interval time where the neuron is quiescent. This implies a potential computational advantage, as well as eliminating one source of estimation variability.

These factors all point to the advantages of estimating TE for event-based data using the continuous-time formalism in Eqs (3) and (4). This paper presents an empirical approach to performing such estimation. The estimator (presented in Methods) operates by considering the probability densities of the history embeddings observed at events and contrasts these with the probability densities of those embeddings being observed at other (randomly sampled) points. This approach is distinct in utilising a novel Bayesian inversion on Eq (4) in order to operate on these probability densities of the history embeddings, rather than making a more difficult direct estimation of spike rates. Furthermore, this allows us to utilise $k$-Nearest-Neighbour ($k$NN) estimators for the entropy terms based on these probability densities. These estimators have known advantages of consistency, data efficiency, low sensitivity to parameters and known bias corrections. By combining these entropy estimators, and making use of established bias reduction techniques for combinations of $k$NN estimators, we arrive at our proposed estimator. The resulting estimator is consistent (see Methods) and is demonstrated on

synthetic examples in Results to be substantially superior to estimators based on time discretisation across a number of metrics.

To conclude that there exists non-zero TE (and thus establish conditional dependence) between two processes a suitable hypothesis test is required. This is usually done by creating a surrogate population of processes (or samples of histories) which conform to the null hypothesis of zero TE, or in other words, directed conditional independence of the target spikes from the source. The algorithm which we use should create surrogates which are identically distributed to the original processes (or history samples) if and only if the null hypothesis holds [47]. The historically used method for generating these surrogates was to either shuffle the original source samples or to shift the source process in time. However, this results in surrogates which conform to an incorrect null hypothesis–that the transitions in the target are completely independent of the source histories. That is, they conform to the factorisation $p(X_t, \mathbf{Y}_{<t} \mid \mathbf{X}_{<t}, \boldsymbol{\mathscr{Z}}_{<t}) = p(X_t \mid \mathbf{X}_{<t}, \boldsymbol{\mathscr{Z}}_{<t})p(\mathbf{Y}_{<t})$. In cases where there is a pairwise correlation between the present state of the target and the history of the source, but they are nonetheless conditionally independent, shuffling or time shifting will create surrogates that are not identically distributed to the original history samples. This is despite the fact that the null hypothesis holds. This can result in the estimate of the TE on the original processes being statistically different from those on the surrogate population, leading to the incorrect inference of non-zero TE.

As shown in Results, this can lead to incredibly high false positive rates for conditional dependence in certain settings such as the presence of strong common driver effects. Therefore, in order to have a suitable significance test for use in conjunction with the proposed estimator, we also present (in Methods) an adaptation of a recently proposed local permutation method [48] to our specific case. This adapted scheme produces surrogates which conform to the correct null hypothesis of conditional independence of the present of the target and the source history, given the histories of the target and further conditioning processes. This is the condition that $p(X_t, \mathbf{Y}_{<t} \mid \mathbf{X}_{<t}, \boldsymbol{\mathscr{Z}}_{<t}) = p(X_t \mid \mathbf{X}_{<t}, \boldsymbol{\mathscr{Z}}_{<t})p(\mathbf{Y}_{<t} \mid \mathbf{X}_{<t}, \boldsymbol{\mathscr{Z}}_{<t})$.

It is easy to intuit that the second factorisation is correct by rewriting the discrete-time TE (Eq (2)) as:

$$\dot{\mathbf{T}}_{Y \to X} \quad = \frac{1}{\tau}\sum_{t=1}^{N_T} \ln \frac{p(x_t, \mathbf{y}_{<t} \mid \mathbf{x}_{<t}, \boldsymbol{\varkappa}_{<t})}{p(x_t \mid \mathbf{x}_{<t}, \boldsymbol{\varkappa}_{<t})p(\mathbf{y}_{<t} \mid \mathbf{x}_{<t}, \boldsymbol{\varkappa}_{<t})}. \tag{5}$$

That is, transfer entropy can be readily interpreted as a measure of the difference between the distributions $p(X_t, \mathbf{Y}_{<t} \mid \mathbf{X}_{<t}, \boldsymbol{\mathscr{Z}}_{<t})$ and $p(X_t \mid \mathbf{X}_{<t}, \boldsymbol{\mathscr{Z}}_{<t})p(\mathbf{Y}_{<t} \mid \mathbf{X}_{<t}, \boldsymbol{\mathscr{Z}}_{<t})$.

We show in Results that the combination of the proposed estimator and surrogate generation method is capable of correctly distinguishing between zero and non-zero information flow in difficult cases, such as where the history of the source has a strong pairwise correlation with the occurrence of events in the target, but is nevertheless conditionally independent. The combination of the current state-of-the-art in discrete-time estimation and a traditional method of surrogate generation is shown to be incapable of making this distinction.

Similarly, we demonstrate that the proposed combination is capable of correctly distinguishing between conditional dependence and independence relationships in data taken from a simple circuit of biophysical model neurons inspired by the crustacean stomatogastric ganglion [49]. Despite the presence of strong pairwise correlations, the success of our estimator here contrasts not only with known failure of a related Granger causality estimator, but also our demonstration that the discrete-time estimator is incapable of correctly performing this task.

Our results provide strong impetus for the application of our proposed techniques to investigate information flows in spike-train data recorded from biological neurons. Furthermore, we underline the importance of our correct identification of conditional dependence and independence in these experiments. Whilst functional/effective network inference algorithms using TE estimators such as ours are *not* expected to align with structural networks in general, they would be expected to do so under certain idealised assumptions (e.g. full observability, large sample size, etc., as outlined in Methods 16) implemented in these experiments. As recently discussed by Novelli and Lizier [50], and specifically for spiking neural networks by Das and Fiete [51], inference aligning with underlying structure under such conditions is a crucial validation that the effective network models they infer at scale are readily interpretable. As such, the demonstration of the efficacy of our proposed approach to detecting conditional dependence in small networks here implies that it holds promise for larger scale effective network inference once paired with a suitable (conditional-independence-based) network inference algorithm (e.g. IDTxl as described in [7, 52]).

## Results

The first two subsections here present the results of the continuous-time estimator applied to two different synthetic examples for which the ground truth value of the TE is known. The first example considers independent processes where $\dot{\mathbf{T}}_{Y\to X} = 0$, whilst the second examines coupled point processes with a known, non-zero $\dot{\mathbf{T}}_{Y\to X}$. The continuous-time estimator's performance is also contrasted with that of the discrete-time estimator. The emphasis of these sections is on properties of the estimators in isolation, as opposed to when combined with a statistical test. As such, we focus on the estimators' bias, variance and consistency (see Methods).

The third, fourth and fifth subsections present the results of the combination of the continuous-time estimator and the local permutation surrogate generation scheme applied to two examples: the first two synthetic and the last a biologically plausible model of neural activity. The comparison of the estimates to a population of surrogates produces *p*-values for the null hypothesis of zero TE. Rejection of this null hypothesis and the resulting conclusion of non-zero TE implies a directed statistical dependence. The results are compared to the known connectivity of the studied systems. Whilst we do not expect directed statistical dependence to have a one-to-one correspondence with structural connectivity in general, these experiments are designed under ideal conditions such that they would. This provides important test cases for detection of conditional dependence and independence. These *p*-values could be translated into other metrics such as ROC curves and false-positive rates, but we choose to instead visualise the distributions of the *p*-values themselves. The combination of the discrete-time estimator along with a traditional method for surrogate generation (time shifts) is also applied to these examples for comparison.

### No TE between independent homogeneous poisson processes

The simplest processes on which we can attempt to validate the estimator are independent homogeneous Poisson processes, where the true value of the TE between such processes is zero.

Pairs of independent homogeneous Poisson processes were generated, each with rate $\bar{\lambda} = 1$, and contiguous sequences of $N_X \in \{1 \times 10^2, 1 \times 10^3, 1 \times 10^4, 1 \times 10^5\}$ target events were selected. For the continuous-time estimator, the parameter $N_U$ for the number of placed sample points was varied (see Methods) to check the sensitivity of estimates to this parameter. At

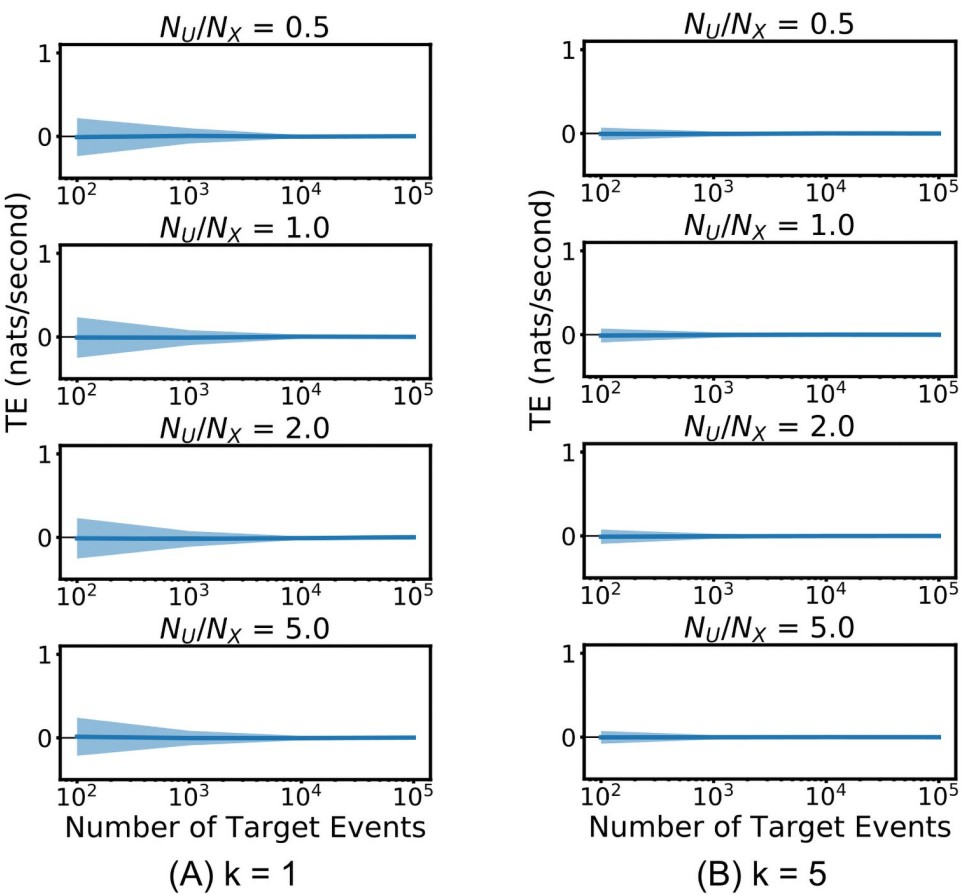

**Fig 2. Evaluation of the continuous-time estimator on independent homogeneous Poisson processes.** The solid line shows the average TE rate across multiple runs and the shaded area spans from one standard deviation below the mean to one standard deviation above it. Plots are shown for two different values of $k$ nearest neighbours, and four different values of the ratio of the number of sample points to the number of events $N_U/N_X$ (See Methods).

each of these numbers of target events $N_X$, the averages are taken across 1000, 100, 20 and 20 tested process pairs respectively.

Fig 2 shows the results of these runs for the continuous-time estimator, using various parameter settings. In all cases, the Manhattan ($\ell_1$) norm is used as the distance metric and the embedding lengths are set to $l_X = l_Y = 1$ spike. See Methods for a description of these parameters. See also [52] for a discussion on how to set these embedding lengths. For this example, the set of conditioning processes $\mathscr{Z}$ is empty. S1 Fig shows results with longer history embeddings.

The plots show that the continuous-time estimator converges to the true value of the TE (equal to 0). This is a numerical confirmation of its consistency for independent processes. Moreover, it exhibits very low bias (as compared to the discrete-time estimator, Fig 3) for all values of the $k$ nearest neighbours and $N_U/N_X$ parameters. The variance is relatively large for $k = 1$, although it dramatically improves for $k = 5$—this reflects known results for variance of this class of estimators as a function of $k$, where generally $k$ above 1 is recommended [53].

Fig 3 shows the result of the discrete-time estimator applied to the same independent homogeneous Poisson processes for two different combinations of the source and target history embedding lengths, $l$ and $m$ time bins, and four different bin sizes $\Delta t$ (see S2 Fig for

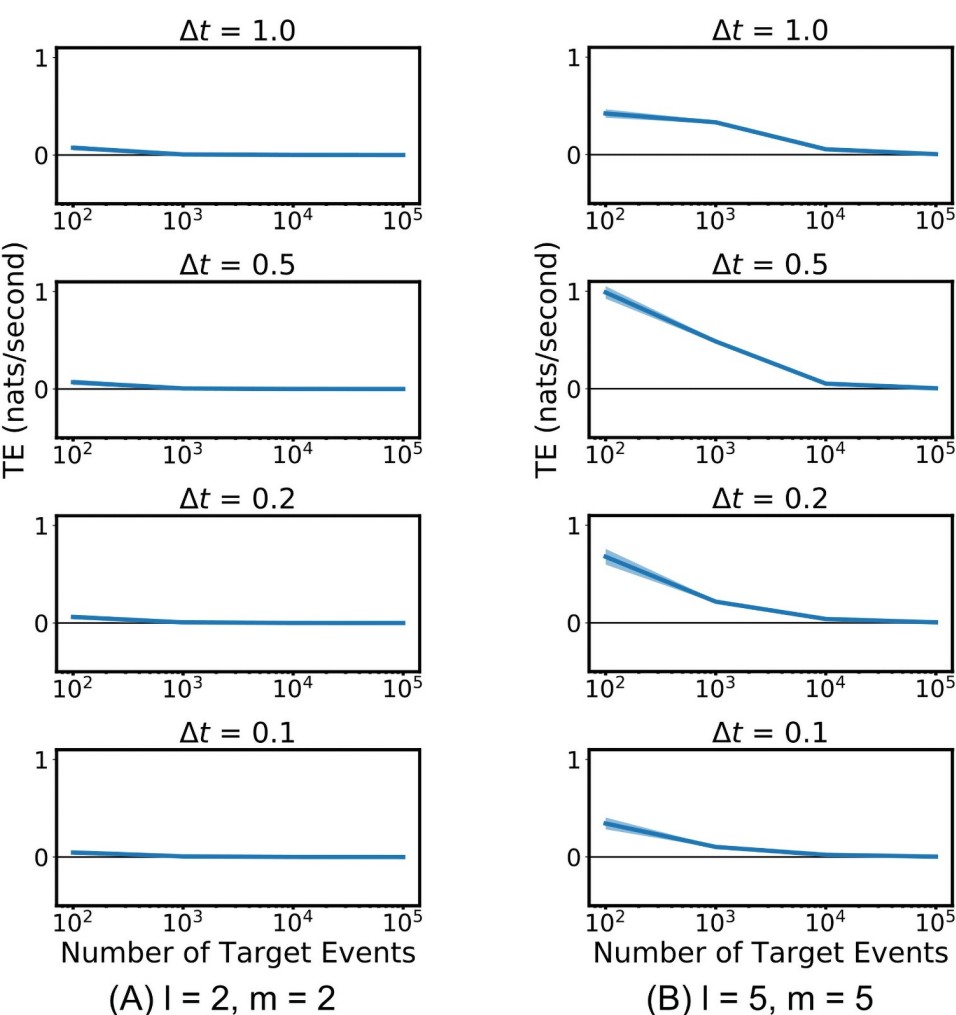

**Fig 3. Result of the discrete-time estimator applied to independent homogeneous Poisson processes.** The solid line shows the average TE rate across multiple runs and the shaded area spans from one standard deviation below the mean to one standard deviation above it. Plots are shown for four different values of the bin width Δt as well as different source and target embedding lengths, *l* and *m*.

different choices of *l* and *m*). At each of the numbers of target events $N_X$, the averages are taken across 1000, 100, 100 and 100 tested process pairs respectively. The variance of this estimator on this process is low and comparable to the continuous-time estimator, however the bias is very large and positive for short processes. The bias of both estimators could be reduced by subtracting the mean of the estimates over a population of surrogates (see the following subsection for an example of this being done with the continuous-time approach). We do observe the discrete-time estimator converging to zero (the true value of the TE) as we increase the available data. This would suggest that it might be consistent on this specific example. However, we will shortly encounter an example where this is not the case.

## Consistent TE between unidirectionally coupled processes

The estimators were also tested on an example of unidirectionally coupled spiking processes with a known value of TE (previously presented as example B in [5]). Here, the source process *Y* is a homogoneous Poisson process. The target process *X* is produced as a conditional point

process where the instantaneous rate is a function of the time since the most recent source event. More specifically:

$$\lambda_{y|\mathbf{x}_{<t},\mathbf{y}_{<t}}[\mathbf{x}_{<t},\mathbf{y}_{<t}] = \bar{\lambda}_y$$

$$\lambda_{x|\mathbf{x}_{<t},\mathbf{y}_{<t}}[\mathbf{x}_{<t},\mathbf{y}_{<t}] = \lambda_x[t_y^1] = \begin{cases} \lambda_x^{\text{base}} & t_y^1 > t_{\text{cut}} \\ \lambda_x^{\text{base}} + m\exp\left[-\frac{1}{2\sigma^2}\left(t_y^1 - \frac{t_{\text{cut}}}{2}\right)^2\right] \\ \quad - m\exp\left[-\frac{1}{2\sigma^2}\left(-\frac{t_{\text{cut}}}{2}\right)^2\right] & t_y^1 \leq t_{\text{cut}}. \end{cases}$$

Here, $t_y^1$ is the time since the most recent source event. As a function of $t_y^1$, the target spike rate $\lambda_{x|\mathbf{x}_{<t},\mathbf{y}_{<t}}[\mathbf{x}_{<t},\mathbf{y}_{<t}]$ rises from a baseline $\lambda_x^{\text{base}}$ at $t_y^1 = 0$ to a peak at $t_y^1 = t_{\text{cut}}/2$, before falling back to the baseline $\lambda_x^{\text{base}}$ from $t_y^1 = t_{\text{cut}}$ onwards (see Fig 4A). We simulated this process using the parameter values $\bar{\lambda}_y = 0.5$, $m = 5$, $t_{\text{cut}} = 1$, $\lambda_x^{\text{base}} = 0.5$ and $\sigma^2 = 0.01$. This simulation was performed using a thinning algorithm [54]. Specifically, we first generated the source process at rate $\bar{\lambda}_y$. We then generate the target as a homogeneous Poisson process with rate $\lambda_h$ such that $\lambda_h > \lambda_x[t_y^1]$ for all values of $t_y^1$. We then went back through all the events in this process and removed each event with probability $1 - \lambda_x[t_y^1]/\lambda_h$. As with the previous example, once a pair of processes had been generated, a contiguous sequence of $N_X$ target events was selected. Tests were conducted for the values of $N_X \in \{1 \times 10^2, 1 \times 10^3, 1 \times 10^4, 1 \times 10^5, 1 \times 10^6\}$. For the continuous-time estimator, the number of placed sample points $N_U$ was set equal to $N_X$ (see Methods). At each $N_X$, the averages are taken over 1000, 100, 20, 20 and 20 tested process pairs respectively.

Spinney et al. [5] present a numerical method for calculating the TE for this process, based on known conditional firing rates in the system under stationary conditions. For the parameter values used here the true value of the TE is 0.5076 ± 0.001.

Given that we know that the dependence of the target on the source is fully determined by the distance to the most recent event in the source, we used a source embedding length of $l_Y = 1$. The estimators were run with three different values of the target embedding length $l_X \in \{1, 2, 3\}$ (see Methods). For this example, the set of conditioning processes $\mathscr{Z}$ is empty.

Fig 4B shows the results of the continuous-time estimator applied to the simulated data. We used the value of $k = 4$ and the Manhattan ($\ell_1$) norm. The results displayed are as expected in that for a short target history embedding length of $l_X = 1$ spike, the estimator converges to a slight over-estimate of the TE. The overestimate at shorter target history embedding lengths $l_X$ can be explained in that perfect estimates of the $\sum_{i=1}^{N_X} \ln \lambda_{x|\mathbf{x}_{<t}}[\mathbf{x}_{<t}]$ component require full knowledge of the target past within the previous $t_{\text{cut}} = 1$ time unit; shorter values of $l_X$ don't cover this period in many cases, leaving this rate underestimated and therefore the TE overestimated. For longer values of $l_X \in \{2, 3\}$ we see that they converge closely to the true value of the TE. This is a further numerical confirmation of the consistency of the continuous-time estimator. See S1 Fig for plots with a different value of $l_Y$.

Fig 4C shows the results of the discrete-time estimator applied to the same process, run for three different values of the bin width $\Delta t \in \{1, 0.5, 0.2, 0.1\}$ time units. The number of bins included in the history embeddings was chosen such that they extended one time unit back (the known length of the history dependence). Smaller bin sizes could not be used as this leads to undersampling of the possible history permutations, resulting in far inferior performance. The plots are a clear demonstration that the discrete-time estimator is very biased and not consistent. At a bin size of $\Delta t = 0.2$ it converges to a value around half the true TE. Moreover, its

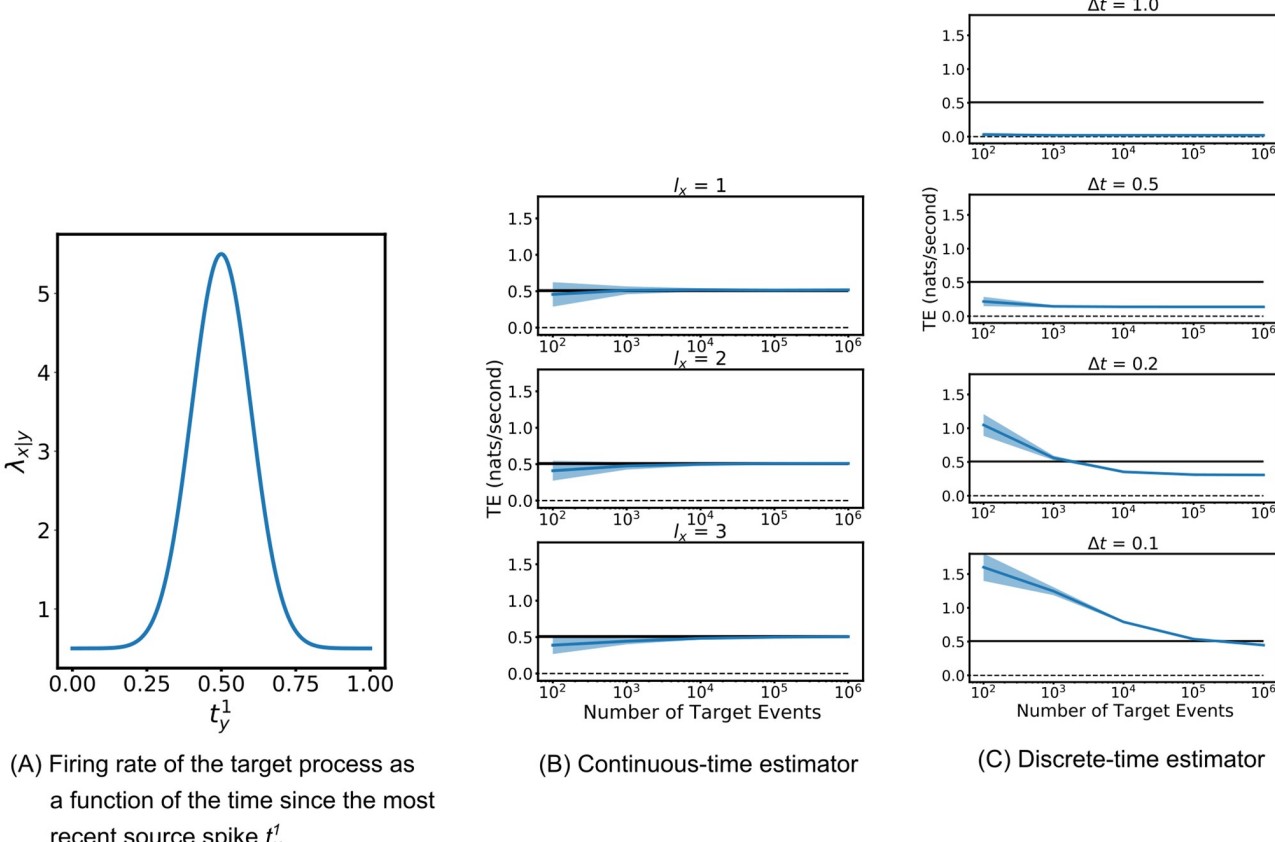

(A) Firing rate of the target process as a function of the time since the most recent source spike $t_y^1$

(B) Continuous-time estimator

(C) Discrete-time estimator

**Fig 4. The discrete-time and continuous-time estimators were run on coupled point processes for which the ground-truth value of the TE is known.** (A) shows the firing rate of the target process as a function of the history of the source. (B) and (C) show the estimates of the TE provided by the two estimators. The solid blue line shows the average TE rate across multiple runs and the shaded area spans from one standard deviation below the mean to one standard deviation above it. The black line shows the true value of the TE. For the continuous-time estimator the parameter values of $N_U/N_X = 1$ and $k = 4$ were used along with the $\ell_1$ (Manhattan) norm. Plots are shown for three different values of the length of the target history component $l_X$. For the discrete-time estimator, plots are shown for four different values of the bin width $\Delta t$. The source and target history embedding lengths are chosen such that they extend back one time unit (the known length of the history dependence).

convergence is incredibly slow. At the bin size of $\Delta t = 0.1$ it would appear to not have converged even after 1 million target events, and indeed it is not even converging to the true value of the TE. The significance of the performance improvement by our estimator is explored further in Discussion.

## Identifying conditional independence despite strong pairwise correlations

The existence of a set of conditioning processes under which the present of the target component is conditionally independent of the past of the source implies that, under certain assumptions, there is no causal connection from the source to the target [55, 56, 57] (see Methods for details on the assumptions we use to conclude the ground truth of dependence/independence in the examples we use here). More importantly, TE can be used to test for such conditional independence (see Methods), thus motivating its use in directed functional (using pairwise TE) and effective (using multivariate TE) network inference. A large challenge faced in testing for conditional independence is correctly identifying "spurious" correlations, whereby conditionally independent components might have a strong pairwise correlation. This problem is particularly pronounced when investigating the spiking activity of biological neurons, as

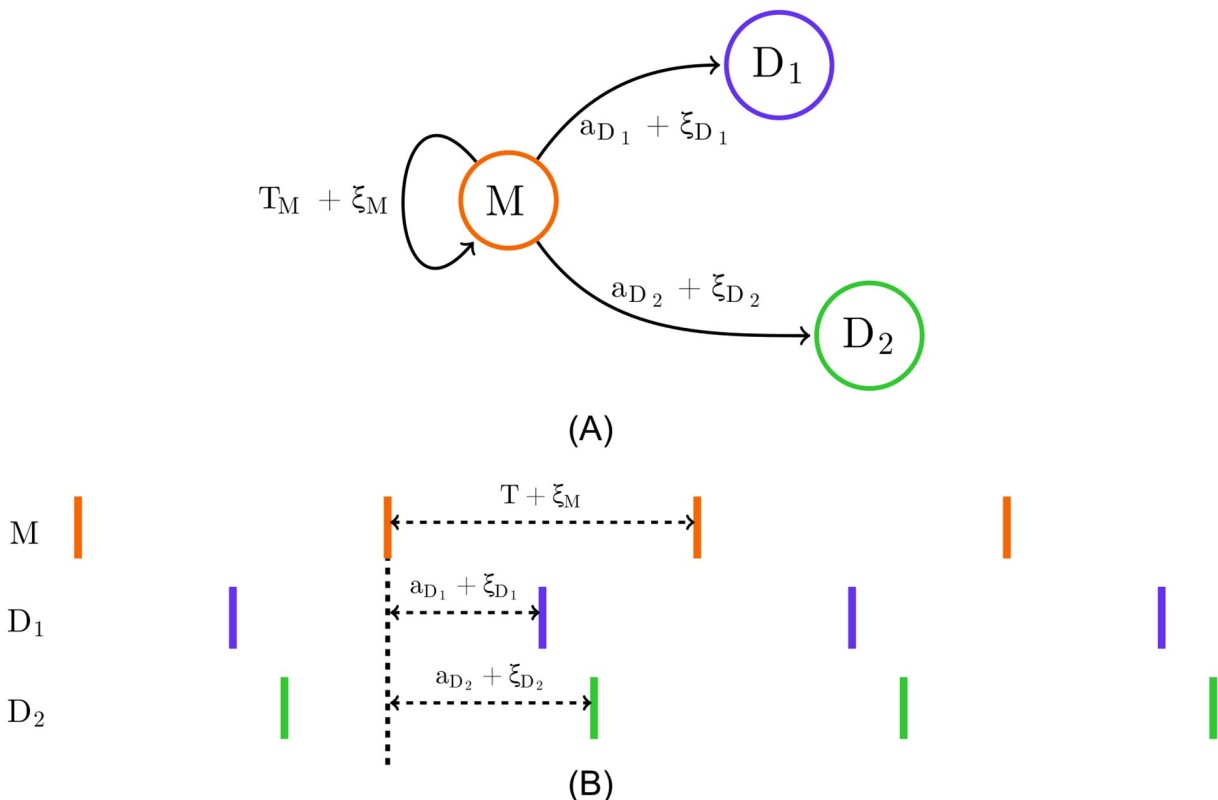

**Fig 5. Diagram of the noisy copy process.** Events in the mother process $M$ occur periodically with intervals $T + \xi_M$ ($\xi_M$ and $\xi_{D_i}$ are noise terms). Events in the daughter processes $D_1$ and $D_2$ occur after each event in the mother process, at a distance of $a_{D_i} + \xi_{D_i}$ (with $a_{D_1} < a_{D_2}$). (A) shows a graph of the dependencies with the labels on the edges representing delays. (B) shows a diagram of a representative spike raster.

populations of which often exhibit highly correlated behaviour through various forms of synchrony [58, 59, 60] or common drivers [61, 62]. In this subsection, we demonstrate that the combination of the presented estimator and surrogate generation scheme is particularly adept at identifying conditional independence in the face of strong pairwise correlations on a synthetic example. Moreover, the combination of the traditional discrete-time estimator and surrogate generation techniques are demonstrated to be ineffective on this task.

The chosen synthetic example in this subsection models a common driver effect, where an apparent directed coupling between a source and target is only due to a common parent. In such cases, despite a strong induced correlation between the source history and the occurrence of an event in the target, we expect to measure zero information flow when conditioning on the common driver. Our system here consists of a quasi-periodic 'mother' process $M$ (the common driver) and two 'daughter' processes, $D_1$ and $D_2$ (see Fig 5A for a diagram of the process). The mother process contains events occurring at intervals of $T + \xi_M$, with the daughter processes being noisy copies with each event shifted by an amount $a_{D_i} + \xi_{D_i}$ ($\xi_M$ and $\xi_{D_i}$ are noise terms). We also choose that $a_{D_1} < a_{D_2}$; so long as the difference between these $a_{D_i}$ values is large compared to the size of the noise terms, this will ensure that the events in $D_1$ precede those in $D_2$. When conditioning on the mother process, the TE from the first daughter to the second, $\dot{T}_{D_1 \to D_2 | M}$, should be 0. However, accurately detecting this is difficult, as the history of source daughter process $D_1$ is strongly correlated with the occurrence of events in the second

daughter process $D_2$—the events in $D_1$ will precede those in $D_2$ by the constant amount $a_{D_2} - a_{D_1}$ plus a small noise term $\xi_{D_2} - \xi_{D_1}$.

Due to the noise in the system, this level of correlation will gradually break down if we translate the source daughter process relative to the others. This allows us to do two things. Firstly, we can get an idea of the bias of the estimator on conditionally independent processes for different levels of pairwise correlation between the history of the source and events in the target. Secondly, we can evaluate different schemes of generating surrogate TE distributions as a function of this correlation. We would expect that, for well-generated surrogates which reflect the relationships to the conditional process, the TE estimates on these conditionally independent processes will closely match the surrogate distribution.

We simulated this process using the parameter values of $T = 1.0$, $a_{D_1} = 0.25$, $a_{D_2} = 0.5$, $\xi_{D_1} \sim \mathcal{N}(0, \sigma_D^2)$ and $\xi_{D_2} \sim \mathcal{N}(0, \sigma_D^2)$ where $\sigma_D = 0.05$. $\xi_M$ was distributed as a left-truncated normal distribution, with mean 0 and standard deviation $\sigma_M = 0.05$, with a left truncation point of $-T + \varepsilon$, where $\varepsilon = 1 \times 10^{-6}$, ensuring that $T + \xi_M > 0$. Once the process had been simulated, the source process $D_1$ was translated by an amount $\omega$. We used values of $\omega$ between -10$T$ and 10$T$, at intervals of 0.13$T$. For each such $\omega$, the process was simulated 200 times. For each simulation, the TE was estimated on the original process with the translation $\omega$ in the first daughter as well as on a surrogate generated according to our proposed local permutation scheme (see Methods for a detailed description). The parameter values of $k_{\mathrm{perm}} = 10$ and $N_{U,\mathrm{surrogate}} = N_X$ were used. For comparison, we also generated surrogates according to the traditional source time-shift method, where this shift was distributed randomly uniform between 200 and 300 time units. A contiguous region of 50 000 target events was extracted and the estimation was performed on this data. The continuous-time estimator used the parameter values of $l_X = l_Y = l_{Z_1} = 1$, $k = 10$, $N_U = N_X$ and the Manhattan ($\ell_1$) norm.

The results in Fig 6A demonstrate that the null distribution of TE values produced by the the local permutation surrogate generation scheme closely matches the distribution of TE values produced by the continuous-time estimator applied to the original data. Whilst the raw TE estimates retain a slight negative bias (explored further in Discussion), we can generate a bias-corrected TE with the surrogate mean subtracted from the original estimate (giving an "effective transfer entropy" [63]). This bias-corrected TE as displayed in Fig 6B is consistent with zero because of the close match between our estimated value and surrogates, which is the desired result in this scenario. On the other hand, the TE values estimated on the surrogates generated by the traditional time-shift method are substantially lower than those estimated on the original process (Fig 6A); comparison to these would produce very high false positive rates for significant directed statistical relationships (see the values of TE bias-corrected to these surrogates, which are not consistent with 0, in Fig 6B). This is most pronounced for high levels of pairwise source-target correlation (with translations $\omega$ near zero). The reason behind this difference in the two approaches is easy to intuit. The traditional time-shift method destroys all relationship between the history of the source and the occurrence of events in the target. This means that we are comparing estimates of the TE on the original processes (where there is a strong pairwise correlation between the history of the source and the occurrence of target events) with estimates of the TE on fully independent surrogate processes. Specifically, in discrete time, the joint distribution of the present state of the target and the source history, conditioned on the other histories decomposes as $p(X_t, \mathbf{Y}_{<t} \mid \mathbf{X}_{<t}, \boldsymbol{\mathcal{Z}}_{<t}) = p(X_t \mid \mathbf{X}_{<t}, \boldsymbol{\mathcal{Z}}_{<t}) p(\mathbf{Y}_{<t})$ when using a naive shift method.

By contrast, the proposed local permutation scheme produces surrogates where, although the history of the source and the occurrence of events in the target are conditionally independent, the relationship between the history of the source and the mediating variable,

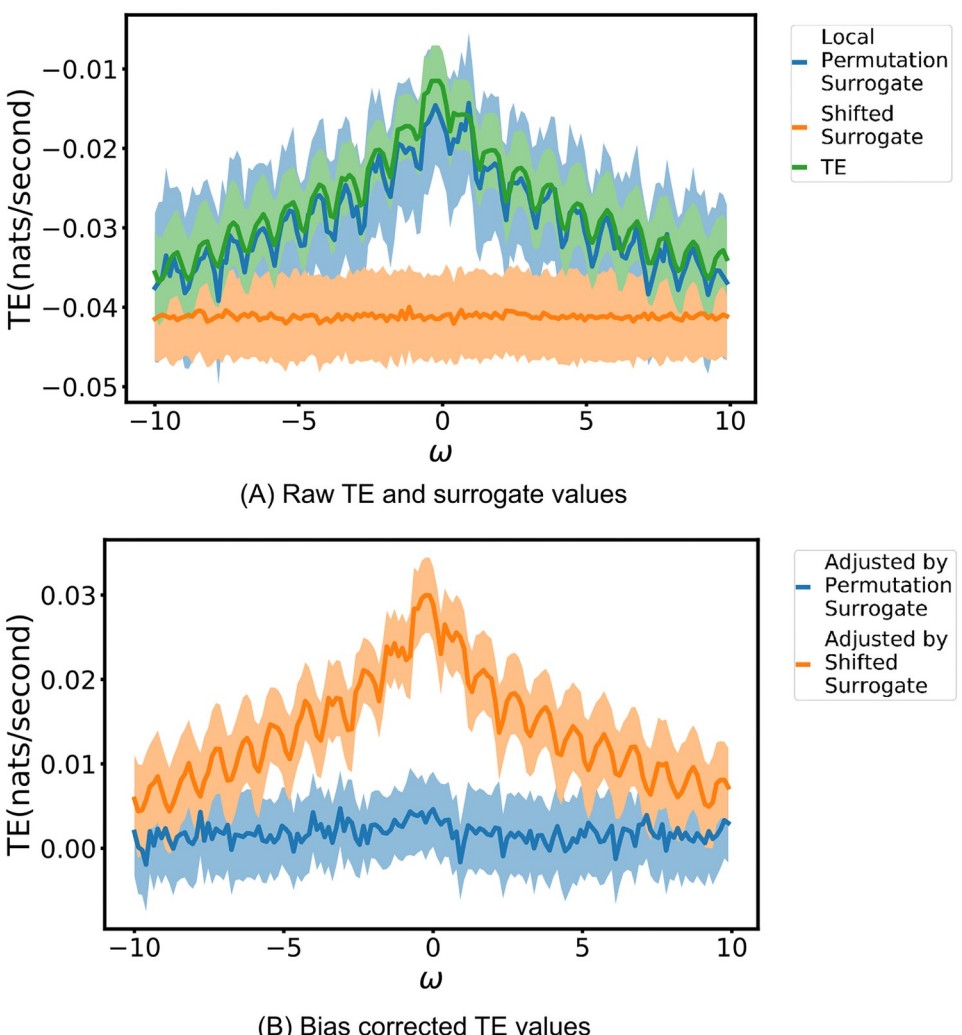

**Fig 6. Results of the continuous-time estimator run on a noisy copy process $\dot{T}_{D_1 \to D_2|M}$, where conditioning on a strong common driver $M$ should lead to zero information flow being inferred.** The translation $\omega$ of the source, relative to the target and common driver, controls the strength of the correlation between the source and target (maximal at zero translation). For each translation, the estimator is run on both the original process as well as embeddings generated via two surrogate generation methods: our proposed local permutation method and a traditional source time-shift method. The solid lines show the average TE rate across multiple runs and the shaded areas span from one standard deviation below the mean to one standard deviation above it. The bias of the estimator changes with the translation $\omega$, and we expect the estimates to be consistent with appropriately generated surrogates reflecting the same strong common driver effect. This is the case for our local permutation surrogates, as shown in (A). This leads to the correct bias-corrected TE value of 0, as shown in (B).

which in this case is the history of the mother process, is maintained. That is, the scheme produces surrogates where (working in the discrete-time formalism for now) the joint distribution of the present of the target and the source history, conditioned on the other histories decomposes appropriately as $p(X_t, \mathbf{Y}_{<t} \mid \mathbf{X}_{<t}, \boldsymbol{\mathcal{Z}}_{<t}) = p(X_t \mid \mathbf{X}_{<t}, \boldsymbol{\mathcal{Z}}_{<t}) p(\mathbf{Y}_{<t} \mid \mathbf{X}_{<t}, \boldsymbol{\mathcal{Z}}_{<t})$. See Methods for the analogous decomposition within the continuous-time event-based TE framework.

We then confirm that the proposed scheme is able to correctly distinguish between cases where an information flow does or does not exist. To do so, we applied it to measure $\dot{\mathbf{T}}_{M \to D_2|D_1}$

in the above system, where we would expect to see non-zero information flow from the common driver or mother to one daughter process, conditioned on the other. The setup used was identical to above however focussing on a translation of $\omega = 0$, and for completeness, two different levels of noise in the daughter processes were used: $\sigma_D = 0.05$ and $\sigma_D = 0.075$. The translation of $\omega = 0$ was chosen as, in the cases of zero information flows ($\dot{\mathbf{T}}_{D_1 \to D_2 | M}$), the pairwise source-target correlations will be at their highest, increasing the difficulty of correctly identifying these zero flows.

We recorded the $p$ values produced by the combination of the proposed continuous-time estimator and the local permutation surrogate generation scheme when testing for conditional information flow where it is expected to be non-zero through $\dot{\mathbf{T}}_{M \to D_2 | D_1}$, in addition to where there is expected to be zero flow through $\dot{\mathbf{T}}_{D_1 \to D_2 | M}$. These flows were measured in 10 runs each and the distributions of the resulting $p$ values are shown in Fig 7. We observe that our proposed combination assigns a $p$ value of zero in every instance of $\dot{\mathbf{T}}_{M \to D_2 | D_1}$ as expected; whilst for $\dot{\mathbf{T}}_{D_1 \to D_2 | M}$ it assigns $p$ values in a broad distribution above zero, meaning the estimates are consistent with the null distribution as expected.

We also applied the combination of the discrete-time estimator and the traditional time-shift method of surrogate generation to this same task of distinguishing between zero and non-zero conditional information flows. We used time bins of width $\Delta t = 0.05$ and history lengths of 7 bins for the target, source and conditional histories. In order to increase the length of history being considered, while keeping the length of the history embeddings constant, application of the discrete-time estimator often makes use of the fact that the present state of the target might be conditionally independent of the most recent source history due to, for instance, transmission delays. In order to exploit this property of the processes, a lag parameter is determined. This lag parameter is a number of time bins to skip between the target present bin and the start of the source history embedding. We followed the current best practice in determining this lag parameter [17]. That is, before calculating the conditional TE from the source to the target, we determined the optimal lag between the conditional history and the target by calculating the pairwise TE between the conditioning process and the target for all lags between 0 and 10. The lag which produced the maximum such TE was used. We then

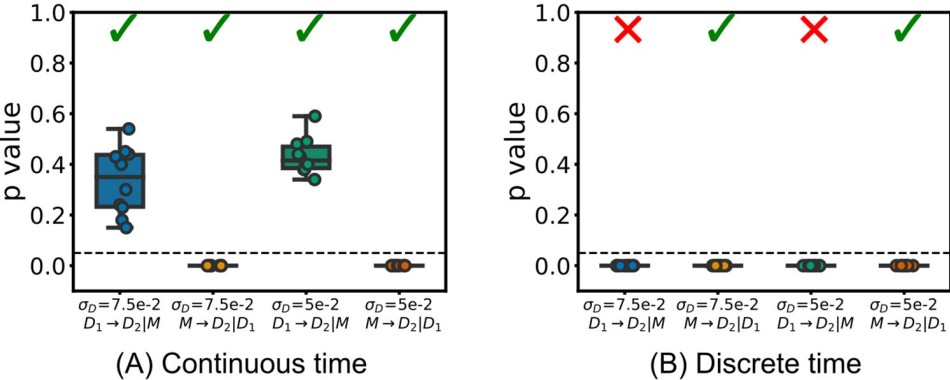

**Fig 7. The $p$-values obtained when using continuous and discrete-time estimators to infer non-zero information flow in the noisy copy process.** The estimators are applied to both $\dot{\mathbf{T}}_{D_1 \to D_2 | M}$ (expected to have zero flow) and $\dot{\mathbf{T}}_{M \to D_2 | D_1}$ (expected to have non-zero flow and therefore be indicated as statistically significant). Only the results from the continuous-time estimator match these expectations. Ticks represent the particular combination of estimator and surrogate generation scheme making the correct inference in the majority of cases when a cutoff value of $p = 0.05$ is used. The dotted line shows $p = 0.05$.

calculated the conditional TE between the source and the target, using this determined lag for the conditioning process, for all lags to the source process between 0 and 10. The TE was then determined to be the maximum TE estimated over all these different lags applied to the source process. This procedure was applied when estimating the TE on the original process as well as on each separate surrogate. The results of this procedure are also displayed in Fig 7. Here we see that the combination of the discrete-time estimator and the traditional time-shift method of surrogate generation assigns a *p* value indistinguishable from zero to all individual runs of both $\dot{\mathbf{T}}_{M \to D_2|D_1}$ and $\dot{\mathbf{T}}_{D_1 \to D_2|M}$. This result–contradicting the expectation that $\dot{\mathbf{T}}_{D_1 \to D_2|M}$ is consistent with zero–suggests that this benchmark approach has an incredibly high false positive rate here.

Finally, we investigated whether the poor performance of the traditional combination of the discrete-time estimator and source time-shift surrogate generation scheme was entirely due to the surrogate generation scheme, or at least partially due to time discretisation. To do so, we reran the experiments for the discrete-time estimator shown in Fig 7B, but replaced the time-shift surrogate generation scheme for an approach which is equivalent to our local permutation scheme, but operates on categorical variables (such as binary numbers). This is a pre-existing conditional-permutation-based surrogate generation technique [64]. The results were identical to those shown in Fig 7B for which the time-shift method of surrogate generation (the usual approach for TE analysis) was used. This suggests that time discretisation plays a substantial role in the failure of the traditional approach on this example. That is, good performance here also requires estimation in continuous time.

## Scaling of conditional independence testing in higher dimensions

The previous subsection demonstrated the ability of the proposed continuous-time TE estimator and local permutation surrogate generation scheme to perform conditional independence tests despite strong pairwise correlations. The results and analysis there demonstrated how the distribution of the TE values over the surrogates was able to match those over the original time series in cases of zero TE, resulting in a broad distribution of *p* values between 0 and 1. It was further demonstrated that the distribution of *p* values obtained from cases with a non-zero TE was clustered around 0, thus providing us with an effective test between zero and non-zero TE. As argued in Introduction and Methods, this is equivalent to a test for conditional independence.

One of the main applications of conditional independence tests is as a component in network inference algorithms [7, 50, 65]. In such cases, the number of processes included in the conditioning set can be as large as one less than the degree of the node. The previous subsection performed a detailed analysis of the distribution of TE values of the original time series, TE values of the surrogate time series as well as the resulting *p* values in a case where there is a single process in the conditioning set. It was also demonstrated that the inference of non-zero TE could be performed successfully in this case. In this subsection, we study the scaling of the inference of non-zero TE with the size of the conditioning set. As such, we provide a demonstration of the suitability of the combination of the proposed estimator and surrogate generation scheme as a component in a conditional-independence based network inference algorithm.

We generate synthetic data on which to test this scaling. The simulated example consists of a single Leaky-Integrate-and-Fire (LIF) [66] neuron and a set of stimuli to it. See Methods for a full description of this model. The LIF neuron has parameters $V_0 = -65$ mv, $V_{\text{reset}} = -75$ mv, $V_{\text{threshold}} = -45$ mv, a time constant of $\tau = 10$ ms and a hard refractory period of 5 ms.

Each stimulus is a separately generated inhomogeneous Poisson process, with an added refractory period of 5 ms. All the stimuli have a common rate. This rate is constant across windows of 0.5$s$ and is generated uniformly randomly between 0 Hz and 40 Hz. As in the above example of unidirectionally coupled process pairs, the stimuli are generated using a thinning algorithm. The process is first generated as a homogeneous Poisson process with rate $R > 40$ Hz. Spikes are excluded with probability $1 - r_i/R$, where $r_i$ is the common rate of the window of the spike. All spikes within the refractory period of the previous spike are also excluded. The stimuli are divided into a set of background processes $B$, with $|B| \in \{6, 12, 18\}$, and a source $Y$. One third of the stimuli in the background set are inhibitory and remainder are excitatory. The strength of the connection $V_{\text{connect}}$ associated with each stimulus was adjusted by hand such that the average firing rate of the target LIF neuron was around 20 Hz when only the stimuli in the background set were connected to the target (that is, the extra source stimulus was unconnected). The resulting connection strengths used are 18 mV, 13 mV and 10 mV for each of the three sizes of the background set, respectively. All connections have a fixed delay of 2 ms. The source stimulus $Y$ is set to be either inhibitory, excitatory or is otherwise unconnected to the target LIF neuron.

In the case where the source neuron is unconnected, when conditioning on all the processes in the background set, the TE between the source and the target LIF neuron is zero. In the cases where it is connected in either an inhibitory or excitatory manner, the TE will be nonzero. This follows from the assumptions made explicit in Methods relating conditional independence and dependence to network structure. We tested the ability of both estimator and surrogate generation scheme combinations to correctly infer zero or non-zero TE.

For the continuous-time estimator and local permutation surrogate generation scheme we used the parameter values of $l_X = l_Y = l_{Z_i} = 1$, $k = 5$, $k_{\text{perm}} = 10$, $N_U/N_X = 1$ and $N_{U,\text{surrogate}}/N_X = 10$. The discrete-time estimator used the same history embedding length for the source, target and conditioning processes. This was set at 3, 2 or 1 bins for each of the conditioning set sizes (6, 12 or 18), respectively. These embedding lengths were chosen so as to keep the total number of bins used across the target, source and conditioning processes below 25. Using more than 25 bins resulted in the space of possible history permutations growing too large, leading to undersampling and far inferior performance. The bin width $\Delta t$ was set at 8 ms, 11 ms and 22 ms for each of these three embedding lengths. These bin widths were chosen so that the history would extend back a distance of at least twice the time constant of the LIF target neuron, plus the transmission delay from the stimuli.

For both combinations, 100 surrogates and a threshold of $p = 0.05$ for the inference of non-zero TE were used. Tests were conducted for the number of target spikes $N_X \in \{100, 500, 1000, 2000, 5000, 10000\}$. For each data set size, both approaches were tested on 30 independent simulations for each setting of $Y$ as either inhibitory, excitatory or unconnected.

Fig 8 shows the results of running the two approaches on the simulated data for different data set sizes. The combination of the discrete-time estimator and the traditional time-shift surrogate generation scheme is found to be inadequate. For data set sizes of $N_X \geq 1000$, we see that this approach assigns non-zero TE to all 30 runs of every connection class (excitatory, inhibitory or absent) at each size, despite the fact that the 30 runs on absent connections correspond to cases of zero TE. Moreover, in the case of absent connections, the direction of convergence is in the wrong direction—this approach performs worse as we provide more data. This is likely due to this scheme's poor ability to identify conditional independence in the presence of pairwise correlations, as we have already seen in the previous subsection. In the instances where the source is not connected to the target, its spiking activity will still be correlated with that of the target, due to it sharing a common rate with the background processes.

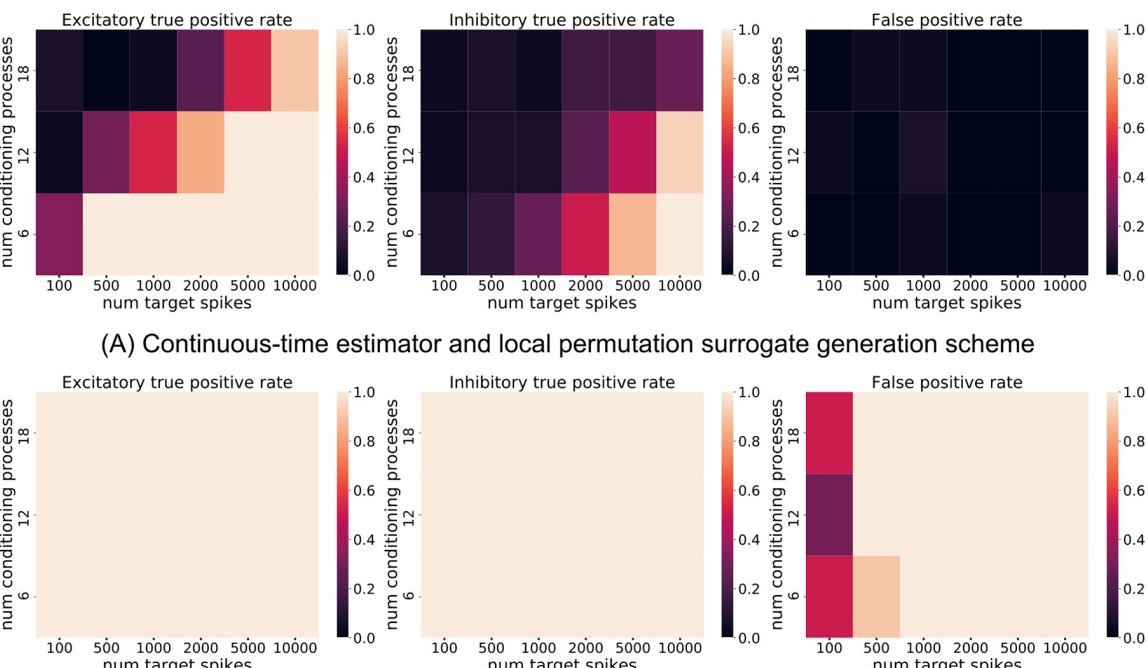

**Fig 8. The scaling of the combination of the continuous-time estimator and the local permutation surrogate generation scheme on correctly identifying conditional independence relationships with increasing dimension and data size (A).** This is compared with the performance of the combination of the discrete-time estimator with the traditional time-shift surrogate generation procedure (B). The *y* axis represents the number of background processes being conditioned on. Above a certain moderate threshold of data size, the discrete-time approach infers a non-zero TE in all of the runs, including those where the source was in fact not connected to the target. This renders it impractical for this task.

S4 Fig displays the same results as Fig 8, but in the simpler case where all stimuli have a constant rate of 20 Hz. In this case, with the pairwise correlations removed, we see that the discrete-time estimator is capable of more consistently correctly identifying cases of zero TE, although it still displays a substantially inflated false positive rate compared to the expected value of 0.05. Moreover, it is worth emphasising that this is an unrealistic scenario as it is assuming completely independent sources, whereas the activity of biological neurons are known to exhibit a wide variety of correlations in their activities [58, 59, 60].

Returning to Fig 8A, in the cases where the conditioning set contains 6 or 12 processes, the combination of the continuous-time estimator and local permutation surrogate generation scheme is able to correctly identify zero versus non-zero TE provided that it has access to around 10000 target spikes. In the case where the conditioning set contains 18 processes, it is capable of correctly identifying non-zero TE for excitatory connections as well as correctly identifying zero TE in the case of an unconnected source. In all combinations of numbers of spikes and number of conditionals, our method is able to control the false positive rate at the prescribed level. This is crucial: in the context of network inference applied to neuroscientific data, false positives are considered more detrimental than false negatives [67]. This is due to such false positives often existing between communities and thus resulting in substantial errors in the inferred topology. With that said, the true positive rate is below 50% for inhibitory sources, though it is observed to rise with an increase in the number of target spikes being considered. Importantly, were a greedy approach to effective network inference to be used, as in

[7, 50], whereby edges are iteratively added to the conditioning set based on their TE value, then the majority of conditional independence tests will be performed at a dimension well below the degree of the node. In order to measure the performance of our proposed approach at the start of this process (where no sources have yet been selected and conditioned on), S5 Fig displays the same results as Fig 8 but where the background processes are not included in the conditioning set. Here we see higher true positive rates at lower numbers of spikes, with the inhibitory connections being easily identified. This implies that, when used as a component in such a greedy algorithm, our approach will be able to identify the principal sources whilst controlling the false-positive rate, although it may miss some true sources in higher dimensions.

Finally, we investigated whether the poor performance of the traditional combination of the discrete-time estimator and source time-shift surrogate generation scheme was entirely due to the surrogate generation scheme. That is, could it be rescued by using a better surrogate generation technique? We therefore repeated the discrete-time experiments shown in Fig 8, S4 and S5 Figs, but replaced the time-shift surrogate generation scheme for an approach which is equivalent to our local permutation scheme, but operates on categorical variables (such as binary numbers). This is an established conditional-permutation based surrogate generation scheme [64]. The results of these runs are displayed in S6 Fig. We observe qualitatively similar results for the use of these two surrogate generation techniques. The only substantial difference is that the conditional-permutation based scheme has lower true positive rates for inhibitory connections when less data is available under all setups. This implies that the poor performance of the traditional approach is largely due to time-discretisation. Once again, we see that good performance here requires estimation in continuous time.

### Testing for conditional independence on the simulated pyloric circuit of the crustacean stomatogastric ganglion

The pyloric circuit of the crustacean stomatogastric ganglion has received significant attention in terms of statistical modelling and has been proposed as a benchmark circuit on which to test spike-based connectivity inference techniques [68, 69]. Such modelling attempts have faced substantial difficulties. For instance, it has been shown that Granger causality is unable to infer the connectivity of this network [68] (Granger causality and TE are equivalent for linear dynamics with Gaussian noise [70]). We demonstrate here that our proposed approach is able to correctly infer the conditional dependence and independence relationships in this circuit (which, as per the previous examples, are expected to match connectivity under the conditions of this experiment, see Methods).

The crustacean stomatogastric ganglion [49, 71, 72] has received substantial research attention as a simple model circuit. The fact that its full connectivity is known is of great use for modelling and statistical analysis. The pyloric circuit is a partially independent component of the greater circuit and consists of an Anterior Burster (AB) neuron, two Pyloric Driver (PD) neurons, a Lateral Pyloric (LP) neuron and multiple Pyloric (PY) neurons. As the AB neuron is electrically coupled to the PD neurons and the PY neurons are identical, for the purposes of modelling, the circuit is usually represented by a single AB/PD complex, and single LP and PY neurons [68, 69, 73, 74].

The AB/PD complex undergoes self-sustained rhythmic bursting. It inhibits the LP and PY neurons through slow cholinergic and fast glutamatergic synapses. These neurons then burst on rebound from this inhibition. The PY and LP neurons also inhibit one another through fast glutamatergic synapses and the LP neuron similarly inhibits the AB/PD complex.

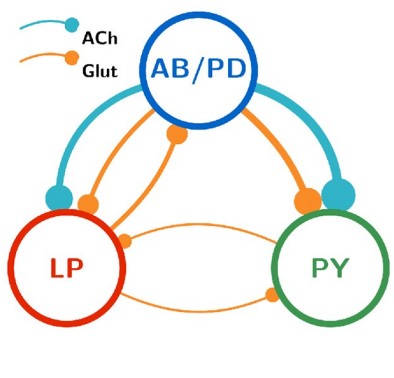

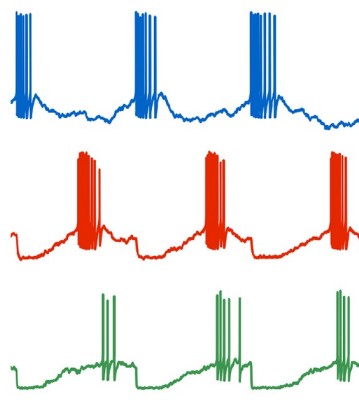

(A) Circuit connectivity diagram

(B) Example membrane potential
traces produced by the circuit.

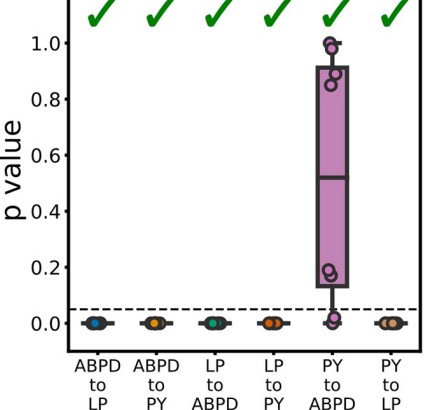

(C) Distribution of *p* values from the
continuous-time estimator and the local
permutation surrogate generation method.

(D) Distribution of *p* values from the
discrete-time estimator and the source
time-shift surrogate generation method

**Fig 9. Results of both estimator and surrogate generation combinations being applied to data from simulations of a biophysical model of a neural circuit inspired by the pyloric circuit of the crustacean stomatogastric ganglion.** The circuit, shown in (A), is fully connected apart from the missing connection between the PY neuron and the AB/PD complex, and generates membrane potential traces which are bursty and highly-periodic with cross-correlated activity. The distribution of *p* values from the combination of the continuous-time estimator and local permutation surrogate generation scheme are shown in (C). They demonstrate that this combination is capable of correctly identifying the conditional dependence and independence relationships in this circuit in all runs, apart from two false negatives. By contrast, the distribution of *p* values produced by the combination of the discrete-time estimator and the traditional source time-shift surrogate generation method shown in (D) mis-specified the relationship from the PY to the ABPD in every run. Ticks represent the particular combination of estimator and surrogate generation scheme making the correct inference of dependence or independence in the majority of cases when a cutoff value of $p = 0.05$ is used.

Fig 9 shows sample membrane potential traces from simulations of this circuit as well as a connectivity diagram. Despite its small size, inference of the relationships between neurons is challenging [68, 69] due to the fact that it is highly periodic. Although there is no structural connection from the PY to the ABPD neuron (implying conditional independence due to full observability and the causal Markov assumption), there is a strong, time-directed, correlation between their activity—the PY neuron always bursts shortly before the ABPD. Recognising

that this is a spurious correlation, and that the AB/PD complex is thus conditionally independent of the history of the PY neuron, requires fully resolving the influence of the AB/PD's history on itself as well as that of the LP on the AB/PD. To further complicate matters, the dependence implied by the connection between the LP and ABPD neurons (along with the contraposition of our assumption of faithfulness) is very challenging to detect. The AB/PD complex will continue bursting regardless of any input from the LP. Correctly inferring this dependence requires detecting the subtle changes in the timing of AB/PD bursts that result from the activity of the LP.

Previous work on statistical modelling of the pyloric circuit has used both *in vitro* and *in silico* data [68, 69]. We ran simulations of biophysical models inspired by this network, similar to those used in [68] (see S1 Text). Attempts were then made to identify the conditional dependence/independence relationships in the network by detecting non-zero conditional information flow from the spiking event times produced by the simulations. This was done by estimating the TE from the source to the target, conditioned on the activity of the third remaining neuron for every source-target pair. Both the combination of the proposed continuous-time estimator and local permutation surrogate generation scheme and the combination of the discrete-time estimator and source time-shift surrogate generation scheme were applied to this task. As the dynamics of the network are fully captured in the three neurons of the network (we have full observability), and due to the causal Markov assumption, in the case where there is no causal directed connection from a source to a target, the target's present will be conditionally independent of the source's past. By the contraposition of the faithfulness assumption, in the presence of a connection the target's current state will be dependent on the source's past (see Methods).

Both combinations were applied to nine independent simulations (ten simulations were instantiated but one was discarded due to early termination from a numerical instability) of the network and the number of target events $N_X = 2 \times 10^4$ was used. For the continuous-time estimator the parameter values of $l_X = l_Y = l_{Z_1} = 3$, $k = 10$, $N_U = N_X$, $N_{U,\text{surrogate}} = 5N_X$ and $k_{\text{perm}} = 10$ were used along with the Manhattan ($\ell_1$) norm (see Methods). The discrete-time estimator made use of a bin size of $\Delta t = 0.05s$ and history embedding lengths of seven bins for each of the source, target and conditioning processes. Searches were performed to determine the optimum embedding lag for both the source and conditioning histories (as above) with a maximum search value of 20 bins being used. We designed the search procedure to include times up to the inter-burst interval (around 1 time unit), which placed an effective lower bound on the width of the time bins (as bin sizes below $\Delta t = 0.05s$ resulted in impractically large search spaces). For both estimators, *p* values were inferred from 100 independently generated surrogates (see Methods). The source time-shift surrogate generation scheme used time shifts distributed uniformly randomly between 200 and 400 time units.

Fig 9C and 9D show the distributions of *p* values resulting from the application of both estimator and surrogate generation scheme combinations. The continuous-time estimator and local permutation surrogate generation scheme were able to correctly infer the dependence/independence relationships in the network in the majority of cases (indicated by *p*-values approaching 0 for the true positives, and spread throughout [0, 1] for the true negatives). On the other hand, the discrete-time estimator and source time-shift surrogate generation scheme produced an erroneous inference on every run: a dependence between the PY neuron and the AB/PD complex. S7 and S8 Figs contain plots showing runs of the continuous-time estimator using different values of the parameters $l_X$, $l_Y$, $l_{Z_1}$ and $N_X$. The results are qualitatively very

similar to those presented in Fig 9C, showing that, on this example, our methodology is robust to these parameter choices.

As in the previous subsections, we investigated whether the poor performance of the traditional combination of the discrete-time estimator and source time-shift surrogate generation scheme was entirely due to the surrogate generation scheme, or at least partially due to time discretisation. To do so, we reran the experiments for the discrete-time estimator shown in Fig 9D, but replaced the time-shift surrogate generation scheme for an approach which is equivalent to our local permutation scheme, but operates on categorical variables (such as binary numbers). As previously, this is a pre-existing conditional-permutation-based surrogate generation method [64]. The results were identical to those shown in Fig 9D for which the time-shift method of surrogate generation (the usual approach for TE analysis) was used. This suggests that time discretisation plays a substantial role in the failure of the traditional approach on this example. Mirroring our previous findings, we observe that good performance here requires estimation in continuous time.

On this particular example, the inference of all connections using the continuous-time approach took 13 minutes and 6 seconds when using 20 cores of an Intel Xeon E5-2670. The discrete-time approach took around 37 minutes and 4 seconds when running on the same hardware. We would, however, point out that the computational requirements for both methods are highly sensitive to their parameters. The discrete-time approach will be particularly sensitive to $\Delta t$ and the number of lag settings searched over. The continuous-time approach is particularly sensitive to the embedding lengths.

## Discussion

Despite transfer entropy being a popular tool within neuroscience and other domains of enquiry [7, 8, 9, 13, 14, 15, 16, 17, 18, 19], it has received more limited application to event-based data such as spike trains. This is at least partially due to current estimation techniques requiring the process to be recast as a discrete-time phenomenon. The resulting discrete-time estimation task has been beset by difficulties including a lack of consistency, high bias, slow convergence and an inability to capture effects which occur over fine and large time scales simultaneously.

This paper has built on recent work presenting a continuous-time formalism for TE [5] in order to derive an estimation framework for TE on event-based data in continuous time. This framework has the unique advantage of only estimating quantities at events in the target process alongside efficient representation of the data as inter-spike intervals, providing a significant computational advantage. Instead of comparing spike rates conditioned on specific histories at each target spiking event, we use a Bayesian inversion to instead make the empirically easier comparison of probabilities of histories at target events versus anywhere else along the target process. This comparison, using KL divergences, is made using $k$-NN techniques, which brings desirable properties such as efficiency for the estimator. This estimator is provably consistent. Moreover, as it operates on inter-event intervals, it is capable of capturing relationships which occur with fine time precision along with those that occur over longer time distances.

The estimator was first evaluated on two simple examples for which the ground truth is known: pairs of independent Poisson processes (first subsection of Results) as well as pairs of processes unidirectionally coupled through a simple functional relationship (second subsection of Results). The current state-of-the-art in discrete-time estimation was also applied to these processes. It was found that the continuous-time estimator had substantially lower bias than the discrete-time estimator, converged orders of magnitude faster (in terms of the

number of sample spikes required), and was relatively insensitive to parameter selections. Moreover, these examples provided numerical confirmation of the consistency of the continuous-time estimator, and further demonstration that the discrete-time estimator is not consistent. The latter simple example highlighted the magnitude of the shortcomings of the discrete-time estimator. In the authors' experience, spike-train datasets which contain 1 million spiking events for a single neuron are vanishingly rare. However, even in the unlikely circumstance that the discrete-time estimator is presented with a dataset of this size, as in the second subsection of Results, it could not accurately estimate the TE for a simple one-way relationship between only two neurons. Moreover, this example neatly demonstrates a known [31], notable problem with the use of the discrete-time estimator, which is that it provides wildly different estimates for different values of $\Delta t$. Whilst the underlying theory [5] suggests that in principle taking the discrete time TE rate as $\Delta t \rightarrow 0$ converges with the continuous time formalism, the use of smaller $\Delta t$ values leads to issues in undersampling and inability to represent patterns on long time scales. In real-world applications, where the ground truth is unknown, there is no principled method for choosing which resulting TE value from the various bin sizes to use.

One of the principal use-cases of TE is the inference of non-zero information flow. As the TE is estimated from finite data, we require a manner of determining the statistical significance of the estimated values. Traditional methods of surrogate generation for TE either shift the source in time, or shuffle the source embeddings. However, whilst this retains the relationship of the target to its past and other conditionals, it completely destroys the relationship between the source and any conditioning processes, which can lead to very high false positive rates as detailed in the third subsection of Results and Methods. We developed a local permutation scheme, based on [48], for use in conjunction with this estimator which is able to maintain the relationship of the source history embeddings with the history embeddings of the target and conditioning processes. The combination of the proposed estimator and this surrogate generation scheme were applied to an example where the history of the source and the occurrence of events in the target are highly correlated, but conditionally independent given their common driver (third subsection of Results). The established time-shift method for surrogate generation produced a null distribution of TE values substantially below that estimated on the original data, incorrectly implying non-zero information flow. Conversely, the proposed local permutation method produced a null distribution which closely tracked the estimates on the original data. The proposed combination was also shown to be able to correctly distinguish between cases of zero and non-zero information flow. When applied to the same example, the combination of the discrete-time estimator and the traditional method of time-shifted surrogates inferred the existence of information flow in all cases, even when no such flow was present. The scaling of these results with the size of the conditioning set was investigated in the fourth subsection of Results. Here, in a highly simplified model of the input-output relationships of a neuron, it was demonstrated that the proposed method could correctly identify conditional dependence vs. independence in cases of up to 12 conditioning processes with access to $10^4$ target spikes. Moreover, it maintained robustness to pairwise correlations despite conditional independence. Again, the traditional combination of discrete-time estimator and time shifted surrogates was found to be lacking.

Finally, our proposed approach was applied to inferring the dependence/independence relationships in a more biologically faithful example in the fifth subsection of Results. For this purpose, we made use of models inspired by the pyloric circuit of the crustacean stomatogastric ganglion. The full observability and large noise provided by this model allowed us to conclude that the conditional dependence/independence relationships would match the underlying connectivity of the model, thus providing us with a ground truth against which to

test our approach. Statistical modelling of this network is challenging due to its highly periodic dynamics. For instance, attempts to use Granger causality, using a more established estimator, to infer its connectivity have been unsuccessful [68]; furthermore, we showed that the discrete-time binary-valued TE estimator (with time-shifted surrogates) also could not successfully infer the independence and dependence relationships in the network. It is worth highlighting in this context that Granger causality and TE are equivalent for linear dynamics with Gaussian noise [70]. Given that discrete-time TE (capable of capturing nonlinear relationships) failed on this network, we suspect that the reason for the earlier failures of Granger causality applied to this network were due, at least in part, to time binning and not entirely due to its inability to find nonlinear relationships. Despite these challenges, our combination of continuous-time estimator and surrogate generation scheme was able to correctly infer the relationships implied by the pyloric network. This provides an important validation of the efficacy of our presented approach on a challenging example of representative biological spiking data.

This work represents a substantial step forward in the estimation of information flows from event-based data. To the best of the authors' knowledge it is the first consistent estimator of TE for event-based data. That is, it is the first estimator which is known to converge to the true value of the TE in the limit of infinite data, let alone to provide efficient estimates with finite data. As demonstrated in the first and second subsections of Results it has substantially favourable bias and convergence properties as compared to the discrete-time estimator. The fact that this estimator uses raw inter-event intervals as its history representation allows it to efficiently capture relevant information from the past of the source, target and conditional processes. This allows it to simultaneously measure relationships that occur both with very fine time scales as well as those that occur over long intervals. This was highlighted in the fifth subsection of Results, where it was shown that our proposed approach is able to correctly infer the conditional dependence/independence relationships implied by a model inspired by the pyloric circuit of the crustacean stomatogastric ganglion. The inference of these relationships requires capturing subtle changes in spike timing. However, its bursty nature means that there are long intervals of no spiking activity. This is contrasted with the poor performance of the discrete-time estimator on this same task, as above. The use of the discrete-time estimator requires a hard trade-off in the choice of bin size: small bins will be able to capture relationships that occur over finer timescales but will result in an estimator that is blind to history effects existing over large intervals. Conversely, whilst larger bins might be capable of capturing these relationships occurring over larger intervals, the estimator will be blind to effects occurring with fine temporal precision.

Further, real-world data is of course sampled at some limited resolution; this means that any estimator cannot detect TE in the underlying process associated with smaller time scales than available in the data, though the consistency property of our continuous-time estimator means that it will converge to the TE value of the process at the available resolution. Of course, as per our Introduction, where temporal resolution in recordings is very poor (such as in calcium imaging experiments) the aforementioned trade-offs for the discrete-time estimator are likely to be less problematic and the advantages of the continuous-time estimator less pronounced.

To the best of our knowledge, this work showcases the first use of a surrogate generation scheme for statistical significance estimates which correctly handles strong source-conditional relationships for event-based data. This has crucial practical benefit in that it greatly reduces the occurrence of false positives in cases where the history of a source is strongly correlated with the present of the target, but conditionally independent.

We make note of the fact that inspection of some plots, notably Fig 6 shows that, in some cases, the estimator can exhibit small though not insignificant bias. Indeed, similar biases can readily be demonstrated with the standard KSG estimator for transfer entropy on continuous

variables in discrete time, in similar circumstances where a strong source-target relationship is fully explained by a conditional process. The reason for the small remaining bias is that while the underlying assumption of the nearest neighbour estimators is of a uniform probability density within the range of the $k$ nearest neighbours, strong conditional relationships tend to result in correlations remaining between the variables within this range. For the common use-case of inferring non-zero information flows this small remaining bias will not be an issue as the proposed method for surrogate generation is capable of producing null distributions with very similar bias properties. Furthermore, such bias can be removed from an estimate by subtracting the mean of the surrogate distribution (as shown via the effective transfer entropy [63] in the third subsection of Results). However, it is foreseeable that certain scenarios might benefit from an estimator with lower bias, without having to resort to generating surrogates. In such cases it will likely prove beneficial to explore the combination of various existing bias reduction techniques for $k$-NN estimators with the approach proposed here. These include performing a whitening transformation on the data [75], transforming each marginal distribution to uniform or exploring alternative approaches to sharing radii across entropy terms (see Methods). The authors believe that the most probable cause of the observed bias in the case of strong pairwise correlations is that these correlations cause the assumption of local uniformity (see Methods) to be violated. Gao, Ver Steeg and Galstyan [76] have proposed a method for reducing the bias of $k$-NN information theoretic estimators which specifically addresses cases where local uniformity does not apply. The application of this technique to our estimator holds promise for addressing this remaining bias.

We foresee that one of the more useful applications of the conditional independence test that the combination of estimator and surrogate generation scheme provides will be network inference. Strictly speaking, statistical methods such as these produce effective network models which are not generally expected to provide precise matches to underlying structural connectivity. Under certain idealised circumstances though, as implemented in our experiments (see Methods), the two can be expected to match, and this provided for the important validation that our methods detect directed conditional independence where it exists in these small networks. The extent to which our method can be validated in this manner on larger more latent-confounded networks, and more importantly the extent to which the network models it infers correlate with underlying structure outside of such idealised conditions including faithfulness (see Methods), remain open questions. This is an intended focus of future work. Indeed, the inference of the connectivity of spiking neural networks from their activity is an active area of research [77, 78] which includes recently proposed continuous-time approaches [79, 80]. However, any conditional independence test will suffer from the curse of dimensionality. This means that performing effective network inference requires pairing the conditional independence test with a suitable (conditional-independence-based) network inference algorithm which reduces the dimensionality of the tests. Fortunately, a variety of such algorithms exist [65] (see Runge [81] for a methodology for reducing the dimensionality outside of network inference). In particular, the greedy algorithm [7, 50], which has already been validated for use in combination with TE (for different types of dynamics on larger networks), holds particular promise. Further, it was recently shown by Das and Fiete [51] that popular existing approaches to the inference of spiking neural networks, such as generalised linear models and maximum entropy-based reverse Ising inference, had very high false-positive rates in instances where the activity of unconnected neurons was highly correlated. Given our focus on demonstrating that our conditional independence test is highly robust to strong pairwise correlations despite conditional independence, we believe that the work presented in this paper holds great promise towards making progress on this important issue.

Finally, it is worth pointing out that, as well as presenting a specific estimator and surrogate generation algorithm, this paper is also presenting an approach to testing for time-directed statistical dependence in spike trains much more generally. Any estimator of KL divergence can be plugged into our framework by being applied to estimate the two KL divergence terms appearing in Eq (10). Moreover, a different surrogate generation scheme could be used, so long as it factorises the distribution of histories as specified in Eq (20) (see Methods). There has been substantial recent progress towards the efficient estimation of divergences [82, 83] in high dimension, pointing to the future promise of this work being applied in the context of network inference.

## Methods

There are a variety of approaches available for estimating information theoretic quantities from continuous-valued data [84]; here we focus on methods for generating estimates $\hat{\dot{\mathbf{T}}}_{Y \to X | \pmb{\mathscr{Z}}}$ of a true underlying (conditional) transfer entropy $\dot{\mathbf{T}}_{Y \to X | \pmb{\mathscr{Z}}}$.

The nature of estimation means that our estimates $\hat{\dot{\mathbf{T}}}_{Y \to X | \pmb{\mathscr{Z}}}$ may have a *bias* with respect to the true value $\dot{\mathbf{T}}_{Y \to X | \pmb{\mathscr{Z}}}$, and a *variance*, as a function of some metric $n$ of the size of the data being provided to the estimator (we use the number of spikes, or events, in the target process). The bias is a measure of the degree to which the estimator systematically deviates from the true value of the quantity being estimated, for finite data size. It is expressed as $\text{bias}(\hat{\dot{\mathbf{T}}}_{Y \to X | \pmb{\mathscr{Z}}}) = \mathbb{E}[\hat{\dot{\mathbf{T}}}_{Y \to X | \pmb{\mathscr{Z}}}] - \dot{\mathbf{T}}_{Y \to X | \pmb{\mathscr{Z}}}$. The variance of an estimator is a measure of the degree to which it provides different estimates for distinct, finite, samples from the same process. It is expressed as $\text{variance}(\hat{\dot{\mathbf{T}}}_{Y \to X | \pmb{\mathscr{Z}}}) = \mathbb{E}[\hat{\dot{\mathbf{T}}}^2_{Y \to X | \pmb{\mathscr{Z}}}] - \mathbb{E}[\hat{\dot{\mathbf{T}}}_{Y \to X | \pmb{\mathscr{Z}}}]^2$. Another important property is *consistency*, which refers to whether, in the limit of infinite data points, the estimator converges to the true value. That is, an estimator is consistent if and only if $\lim_{n \to \infty} \hat{\dot{\mathbf{T}}}_{Y \to X | \pmb{\mathscr{Z}}} = \dot{\mathbf{T}}_{Y \to X | \pmb{\mathscr{Z}}}$.

The first half of this methods section is concerned with the derivation of a consistent estimator of TE which operates in continuous time. In order to be able to test for non-zero information flow given finite data, we require a surrogate generation scheme to use in conjunction with the estimator. Such a surrogate generation scheme should produce surrogate history samples that conform to the null hypothesis of zero information flow. The second half of this section will focus on a scheme for generating these surrogates.

The presented estimator and surrogate generation scheme have been implemented in a software package which is freely available online (see the Implementation subsection).

### Continuous-time estimator for transfer entropy between spike trains

In the following subsections, we describe the algorithm for our estimator $\hat{\dot{\mathbf{T}}}_{Y \to X | \pmb{\mathscr{Z}}}$ for the transfer entropy between spike trains. We first outline our choice of a *k*NN type estimator, due to the desirable consistency and bias properties of this class of estimator. In order to use such an estimator type, we then describe a Bayesian inversion we apply to the definition of transfer entropy for spiking processes, which allows us to operate on probability densities of histories of the processes, rather than directly on spike rates. This results in a sum of differential entropies to which *k*NN estimator techniques can be applied. The evaluation of these entropy terms using *k*NN estimators requires a method for sampling history embeddings, which is presented before attention is turned to a technique for combining the separate *k*NN estimators in a manner that will reduce the bias of the final estimate.

**Consideration of estimator type.** Although there has been much recent progress on parametric information-theoretic estimators [85], such estimators will always inject modelling assumptions into the estimation process. Even in the case that large, general, parametric models are used—as in [82]—there are no known methods of determining whether such a model is capturing all dependencies present within the data.

In comparison, nonparametric estimators make less explicit model assumptions regarding the probability distributions. Early approaches included the use of kernels for the estimation of the probability densities [86], however this has the disadvantage of operating at a fixed kernel 'resolution'. An improvement was achieved by the successful, widely applied, class of nonparametric estimators making use of *k*-nearest-neighbour statistics [53, 87, 88, 89], which dynamically adjust their resolution given the local density of points. Crucially, there are consistency proofs [88, 90] for *k*NN estimators, meaning that these methods are known to converge to the true values in the limit of infinite data size. These estimators operate by decomposing the information quantity of interest into a sum of differential entropy terms $H^*$. Each entropy term is subsequently estimated by estimating the probability densities $p(x_i)$ at all the points in the sample by finding the distances to the *k*th nearest neighbours of the points $x_i$. The average of the logarithms of these densities is found and is adjusted by bias correction terms. In some instances, most notably the Kraskov-Stögbauer-Grassberger (KSG) estimator for mutual information [53], many of the terms in each entropy estimate cancel and so each entropy is only implicitly estimated.

Such bias and consistency properties are highly desirable–given the efficacy of *k*NN estimators, it would be advantageous to be able to make use of such techniques in order to estimate the transfer entropy of point processes in continuous time. However the continuous time formulations in Eqs (3) and (4) contain no entropy terms, being written in terms of *rates* as opposed to probability densities. Moreover, the estimators for each differential entropy term $H^*$ in a standard *k*NN approach operate on sets of points in $\mathbb{R}^d$, and it is unclear how to sample points so as to get an unbiased estimate of the rate.

The following subsection is concerned with deriving an expression for continuous-time transfer entropy on spike trains as a sum of $H^*$ terms, in order to define a *k*NN type estimator.

**Formulating continuous-time TE as a sum of differential entropies.** Consider two point processes $X$ and $Y$ represented by sets of real numbers, where each element represents the time of an event. That is, $X \in \mathbb{R}^{N_X}$ and $Y \in \mathbb{R}^{N_Y}$. Further, consider the set of extra conditioning point processes $\mathscr{Z} = \{Z_1, Z_2, \ldots, Z_{n_{\mathscr{Z}}}\}, Z_i \in \mathbb{R}^{N_{Z_i}}$. We can define a *counting process* $\mathbf{N}_X(t)$ on $X$. $\mathbf{N}_X(t)$ is a natural number representing the 'state' of the process. This state is incremented by one at the occurrence of an event. The instantaneous firing rate of the target is then $\lambda_X(t) = \lim_{\Delta t \to 0} p(\mathbf{N}_X(T + \Delta t) - \mathbf{N}_X(t) = 1)/\Delta t$. Using this expression, Eq (4) can then be rewritten as

$$\dot{\mathbf{T}}_{Y \to X | \mathscr{Z}} = \bar{\lambda}_X \lim_{\Delta t \to 0} \mathbb{E}_{P_X} \left[ \ln \frac{p_U(\mathbf{N}_X(x + \Delta t) - \mathbf{N}_X(x) = 1 \mid \mathbf{x}_{<x}, \mathbf{y}_{<x}, \boldsymbol{\varkappa}_{<x})}{p_U(\mathbf{N}_X(x + \Delta t) - \mathbf{N}_X(x) = 1 \mid \mathbf{x}_{<x}, \boldsymbol{\varkappa}_{<x})} \right]. \tag{6}$$

Here, $\bar{\lambda}_X$ is the average, unconditional, firing rate of the target process, that is $\bar{\lambda}_X = \lim_{N_X, \tau \to \infty} N_X/\tau$. In practice this is estimated through a trivial bias free estimate e.g. $\hat{\bar{\lambda}}_X = (N_X - 1)/\tau$ with $\tau = x_{N_X} - x_1$. $\mathbf{x}_{<x} \in \mathbf{X}_{<X}, \mathbf{y}_{<x} \in \mathbf{Y}_{<X}$ and $\boldsymbol{\varkappa}_{<x} = \{\mathbf{z}_{1,<x}, \mathbf{z}_{2,<x}, \ldots, \mathbf{z}_{n_{\varkappa},<x}\} \in \mathscr{Z}_{<X}$ are the histories of the target, source and conditioning processes, respectively, at time $x$. The probability density $p_U$ is taken to represent the probability density at any arbitrary point in the target process, unconditional of events in any of the

processes. Conversely, $p_X$ is taken to represent the probability density of observing a quantity at target events. The expectation $\mathbb{E}_{P_X}$ is taken over this distribution. That is $\mathbb{E}_{P_X}[f(Y)] = \int_Y f(y)p_X(y)dy$.

By applying Bayes' rule we can make a Bayesian inversion to arrive at:

$$
\begin{aligned}
\dot{\mathbf{T}}_{Y \to X|\mathcal{Z}} = \bar{\lambda}_X \lim_{\Delta t \to 0} \mathbb{E}_{P_X} & \left[ \ln \frac{p_U(\mathbf{x}_{<x}, \mathbf{y}_{<x}, \varkappa_{<x} \mid \mathbf{N}_X(x + \Delta t) - \mathbf{N}_X(x) = 1)}{p_U(\mathbf{x}_{<x}, \varkappa_{<x} \mid \mathbf{N}_X(x + \Delta t) - \mathbf{N}_X(x) = 1)} \right. \\
& \left. \times \frac{p_U(\mathbf{x}_{<x}, \varkappa_{<x})}{p_U(\mathbf{x}_{<x}, \mathbf{y}_{<x}, \varkappa_{<x})} \right].
\end{aligned}
\tag{7}
$$

Eq (7) can be written as

$$
\dot{\mathbf{T}}_{Y \to X|\mathcal{Z}} = \bar{\lambda}_X \mathbb{E}_{P_X} \left[ \ln \frac{p_X(\mathbf{x}_{<x}, \mathbf{y}_{<x}, \varkappa_{<x})}{p_X(\mathbf{x}_{<x}, \varkappa_{<x})} + \ln \frac{p_U(\mathbf{x}_{<x}, \varkappa_{<x})}{p_U(\mathbf{x}_{<x}, \mathbf{y}_{<x}, \varkappa_{<x})} \right].
\tag{8}
$$

Eq (8) can be written as a sum of differential entropy and cross entropy terms

$$
\begin{aligned}
\dot{\mathbf{T}}_{Y \to X|\mathcal{Z}} = \bar{\lambda}_X \quad & [-H(\mathbf{X}_{<X}, \mathbf{Y}_{<X}, \mathcal{Z}_{<X}) + H(\mathbf{X}_{<X}, \mathcal{Z}_{<X}) \\
& + H_{P_U}(\mathbf{X}_{<X}, \mathbf{Y}_{<X}, \mathcal{Z}_{<X}) - H_{P_U}(\mathbf{X}_{<X}, \mathcal{Z}_{<X})].
\end{aligned}
\tag{9}
$$

Here, $H$ refers to an entropy term and $H_{P_U}$ refers to a cross entropy term. More specifically,

$$
H(\mathbf{X}_{<X}, \mathcal{Z}_{<X}) = -\int p_X(\mathbf{x}_{<x}, \varkappa_{<x}) \ln p_X(\mathbf{x}_{<x}, \varkappa_{<x}) d\mathbf{x}_{<x} d\varkappa_{<x}
$$

and

$$
H_{P_U}(\mathbf{X}_{<X}, \mathcal{Z}_{<X}) = -\int p_X(\mathbf{x}_{<x}, \varkappa_{<x}) \ln p_U(\mathbf{x}_{<x}, \varkappa_{<x}) d\mathbf{x}_{<x} d\varkappa_{<x}.
$$

It is worth noting in passing that Eq (8) can also be written as a difference of Kullback-Leibler divergences:

$$
\begin{aligned}
\dot{\mathbf{T}}_{Y \to X|\mathcal{Z}} = \bar{\lambda}_X [ & D_{KL}(P_X(\mathbf{X}_{<X}, \mathbf{Y}_{<X}, \mathcal{Z}_{<X}) || P_U(\mathbf{X}_{<X}, \mathbf{Y}_{<X}, \mathcal{Z}_{<X})) \\
& - D_{KL}(P_X(\mathbf{X}_{<X}, \mathcal{Z}_{<X}) || P_U(\mathbf{X}_{<X}, \mathcal{Z}_{<X}))].
\end{aligned}
\tag{10}
$$

The expressions in Eqs (9) and (10) represent a general framework for estimating the TE between point processes in continuous time. Any estimator of differential entropy $\hat{H}$ which can be adapted to the estimation of cross entropies can be plugged into Eq (9) in order to estimate the TE. Similarly, any estimator of the KL divergence can be plugged into Eq (10).

**Constructing $k$NN estimators for differential entropies and cross entropies.** Following similar steps to the derivations in [53, 75, 90], assume that we have an (unknown) probability distribution $\mu(\mathbf{x})$ for $\mathbf{x} \in \mathbb{R}^d$. Note that here $\mathbf{X}$ is a general random variable (not necessarily a point process). We also have a set $X$ of $N_X$ points drawn from $\mu$. In order to estimate the differential entropy $H$ we need to construct estimates of the form

$$
\hat{H}(X) = -\frac{1}{N_X} \sum_{i=1}^{N_X} \ln \widehat{\mu(\mathbf{x}_i)}
\tag{11}
$$

where $\ln \widehat{\mu(\mathbf{x}_i)}$ is an estimate of the logarithm of the true density. Denote by $\epsilon(k, \mathbf{x}_i, X)$ the distance to the $k$th nearest neighbour of $\mathbf{x}_i$ in the set $X$ under some norm $L$. Further, let $p_i^\mu$ be the probability mass of the $\epsilon$-ball surrounding $\mathbf{x}_i$. If we make the assumption that $\mu(\mathbf{x}_i)$ is constant within the $\epsilon$-ball, we have $p_i^\mu = \frac{k}{N_X - 1} = c_{d,L} \epsilon(k, \mathbf{x}_i, X)^d \mu(\mathbf{x}_i)$ where $c_{d,L}$ is the volume of the

$d$-dimensional unit ball under the norm $L$. Using this relationship, we can construct a simple estimator of the differential entropy:

$$\hat{H}(X) = -\frac{1}{N_X}\sum_{i=1}^{N_X}\ln\frac{k}{(N_X-1)c_{d,L}\epsilon(k,\mathbf{x}_i,X)^d}. \qquad (12)$$

We then add the bias-correction term $\ln k - \psi(k)$. $\psi(x) = \Gamma^{-1}(x)d\Gamma(x)/dx$ is the digamma function and $\Gamma(x)$ the gamma function. This yields $\hat{H}_{\mathrm{KL}}$, the Kozachenko-Leonenko [87] estimator of differential entropy:

$$\hat{H}_{\mathrm{KL}}(X) = -\psi(k) + \ln(N_X-1) + \ln c_{d,L} + \frac{d}{N_X}\sum_{i=1}^{N_X}\ln\epsilon(k,\mathbf{x}_i,X). \qquad (13)$$

This estimator has been shown to be consistent [87, 91].

Assume that we now have two (unknown) probability distributions $\mu(\mathbf{x})$ and $\beta(\mathbf{x})$. We have a set $X$ of $N_X$ points drawn from $\mu$ and a set $Y$ of $N_Y$ points drawn from $\beta$. Using similar arguments to above, we denote by $\epsilon(k, \mathbf{x}_i, Y)$ the distance from the $i$th element of $X$ to its $k$th nearest neighbour in $Y$. We then make the assumption that $\beta(\mathbf{x}_i)$ is constant within the $\epsilon$-ball, and we have $p_i^\beta = \frac{k}{N_Y} = c_{d,L}\epsilon(k,\mathbf{x}_i,Y)^d\beta(\mathbf{x}_i)$. We can then construct a naive estimator of the cross entropy

$$\hat{H}_\beta(X) = -\frac{1}{N_X}\sum_{i=1}^{N_X}\ln\frac{k}{N_Y c_{d,L}\epsilon(k,\mathbf{x}_i,Y)^d}. \qquad (14)$$

Again, we add the bias-correction term $\ln k - \psi(k)$ to arrive at an estimator of the cross entropy.

$$\hat{H}_{\beta,\mathrm{KL}}(X) = -\psi(k) + \ln N_Y + \ln c_{d,L} + \frac{d}{N_X}\sum_{i=1}^{N_X}\ln\epsilon(k,\mathbf{x}_i,Y). \qquad (15)$$

This estimator has been shown to be consistent [91].

Attention should be brought to the fundamental difference between estimating entropies and cross entropies using $k$NN estimators. An entropy estimator takes a set $X$ and, for each $x_i \in X$, performs a nearest neighbour search *in the same set $X$*. An estimator of cross entropy takes two sets, $X$ and $Y$ and, for each $x_i \in X$, performs a nearest neighbour search *in the other set $Y$*.

We will be interested in applying these estimators to the entropy and cross entropy terms in Eq (9). For instance, we could use $\hat{H}_{\beta,\mathrm{KL}}(X)$ to estimate $H_{P_U}(\mathbf{X}_{<X}, \boldsymbol{\mathcal{Z}}_{<X})$, where we have that $\mu = p_X(\mathbf{x}_{<x}, \boldsymbol{\varkappa}_{<x})$ and $\beta = p_U(\mathbf{x}_{<x}, \boldsymbol{\varkappa}_{<x})$. This will be covered in more detail in a later subsection, after we first consider how to represent the history embeddings $\mathbf{x}_{<x}, \mathbf{y}_{<x}, \boldsymbol{\varkappa}_{<x}$ as well as sample them from their distributions.

**Selection and representation of sample histories for entropy estimation.** Inspection of Eqs (8) and (9) informs us that we will need to be able to estimate four distinct differential entropy terms and, implicitly, the associated probability densities:

1. The probability density of the target, source and conditioning histories at target events $p_X(\mathbf{x}_{<x}, \mathbf{y}_{<x}, \boldsymbol{\varkappa}_{<x})$.

2. The probability density of the target, and conditioning histories at target events $p_X(\mathbf{x}_{<x}, \boldsymbol{\varkappa}_{<x})$.

3. The probability density of the target, source and conditioning histories independent of target activity $p_U(\mathbf{x}_{<x}, \mathbf{y}_{<x}, \boldsymbol{\varkappa}_{<x})$.

4. The probability density of the target and conditioning histories independent of target activity $p_U(\mathbf{x}_{<x}, \boldsymbol{\varkappa}_{<x})$.

Estimation of these probability densities will require an associated set of samples for a $k$NN estimator to operate on. These samples for $\mathbf{x}_{<x}, \mathbf{y}_{<x}, \boldsymbol{\varkappa}_{<x}$ will logically be represented as history embeddings from the raw event times of the target $X \in \mathbb{R}^{N_X}$, source $Y \in \mathbb{R}^{N_Y}$ and conditioning $\boldsymbol{\mathcal{Z}} = \{Z_1, Z_2, \ldots, Z_{n_{\boldsymbol{\mathcal{Z}}}}\}, Z_i \in \mathbb{R}^{N_{Z_i}}$ processes. It is assumed that these sets are indexed in ascending order (from the first event to the last). The length of the history embeddings (in terms of how many previous spikes are referred to) must be restricted in order to avoid the difficulties associated with the estimation of probability densities in high dimensions. The lengths of the history embeddings along each process are specified by the parameters $l_X, l_Y, l_{Z_1}, \ldots, l_{Z_{n_{\boldsymbol{\mathcal{Z}}}}}$.

We label the sets of samples as $J_{<X} = \{\mathbf{j}_{<x_i}\}_{i=1}^{N_X}$, $C_{<X} = \{\mathbf{c}_{<x_i}\}_{i=1}^{N_X}$, $J_{<U} = \{\mathbf{j}_{<u_i}\}_{i=1}^{N_U}$, and $C_{<U} = \{\mathbf{c}_{<u_i}\}_{i=1}^{N_U}$, for each probability density $p_X(\mathbf{x}_{<x}, \mathbf{y}_{<x}, \boldsymbol{\varkappa}_{<x}), p_X(\mathbf{x}_{<x}, \boldsymbol{\varkappa}_{<x}), p_U(\mathbf{x}_{<x}, \mathbf{y}_{<x}, \boldsymbol{\varkappa}_{<x})$, and $p_U(\mathbf{x}_{<x}, \boldsymbol{\varkappa}_{<x})$ respectively ($J$ for 'joint' and $C$ for 'conditioning', i.e. without the source).

For the two sets of joint embeddings $J_{<*}$ (where $* \in \{X, U\}$) each $\mathbf{j}_{<*_i} \in J_{<*}$ is made up of target, source and conditioning components. That is, $\mathbf{j}_{<*_i} = \{\mathbf{x}_{<*_i}, \mathbf{y}_{<*_i}, \boldsymbol{\varkappa}_{<*_i}\}$ where $\boldsymbol{\varkappa}_{<*_i} = \{\mathbf{z}_{1,<*_i}, \mathbf{z}_{2,<*_i}, \ldots, \mathbf{z}_{n_{\boldsymbol{\varkappa}},<*_i}\}$. Similarly, for the two sets of conditioning embeddings $C_{<*}$ (where $* \in \{X, U\}$) each $\mathbf{c}_{<*_i} \in C_{<*}$ is made up of target, and conditioning components. That is, $\mathbf{c}_{<*_i} = \{\mathbf{x}_{<*_i}, \boldsymbol{\varkappa}_{<*_i}\}$.

Each set of embeddings $J_{<*}$ is constructed from a set of observation points $T \in \mathbb{R}^{N_T}$. Each individual embedding $\mathbf{j}_{<*_i}$ is constructed at one such observation $t_i$. We denote by $\mathrm{pred}(t_i, P)$, the index of the most recent event in the process $P$ to occur before the observation point $t_i$.

The values of $\mathbf{x}_{<*_i} = \{x_{<*_i}^1, x_{<*_i}^2, \ldots, x_{<*_i}^{l_X}\} \in X_{<*}$ are set as follows:

$$x_{<*_i}^k := \begin{cases} t_i - x_{\mathrm{pred}(t_i,X)} & k = 1 \\ x_{\mathrm{pred}(t_i,X)-k+2} - x_{\mathrm{pred}(t_i,X)-k+1} & k \neq 1. \end{cases} \tag{16}$$

Here, the $t_i \in T$ are the raw observation points and the $x_j \in X$ are the raw event times in the process $X$. The first element of $\mathbf{x}_{<*_i}$ is then the interval between the observation time and the most recent target event time $x_{\mathrm{pred}(t_i, X)}$. The second element of $\mathbf{x}_{<*_i}$ is the inter-event interval between this most recent event time and the next most recent event time and so forth. The values of $\mathbf{y}_{<*_i} = \{y_{<*_i}^1, y_{<*_i}^2, \ldots, y_{<*_i}^{l_X}\} \in Y_{<*}$ and $\mathbf{z}_{<*_i} = \{z_{<*_i}^1, z_{<*_i}^2, \ldots, z_{<*_i}^{l_X}\} \in \boldsymbol{\mathcal{Z}}_{<*}$ are set in the same manner.

The set of samples $J_{<X} = \{\mathbf{j}_{<x_i}\}_{i=1}^{N_X} \subseteq \mathbb{R}^{l_X+l_Y+\sum l_{Z_j}}$ for $p_X(\mathbf{x}_{<x}, \mathbf{y}_{<x}, \boldsymbol{\varkappa}_{<x})$ is constructed using this scheme, with the set of observation points $T$ being simply set as the $N_X$ event times $x_j$ of the target process $X$. As such, $J_{<X} = X_{<X} \times Y_{<X} \times \boldsymbol{\mathcal{Z}}_{<X}$.

In contrast, while the set of samples $J_{<U} = \{\mathbf{j}_{<u_i}\}_{i=1}^{N_U} \subseteq \mathbb{R}^{l_X+l_Y+\sum l_{Z_j}}$ for $p_U(\mathbf{x}_{<x}, \mathbf{y}_{<x}, \boldsymbol{\varkappa}_{<x})$ is also constructed using this scheme, the set of observation points $T$ is set as $U \subseteq \mathbb{R}^{N_U}$. $U$ is composed of sample time points placed independently of the occurrence of events in the target process. These $N_U$ sample points between the first and last events of the target process $X$ can either be placed randomly or at fixed intervals. In the experiments presented in this paper they were placed at fixed intervals. Importantly, note that $N_U$ is not necessarily equal to $N_X$, with their ratio $N_U/N_X$ a parameter for the estimator which is investigated in our Results. We also

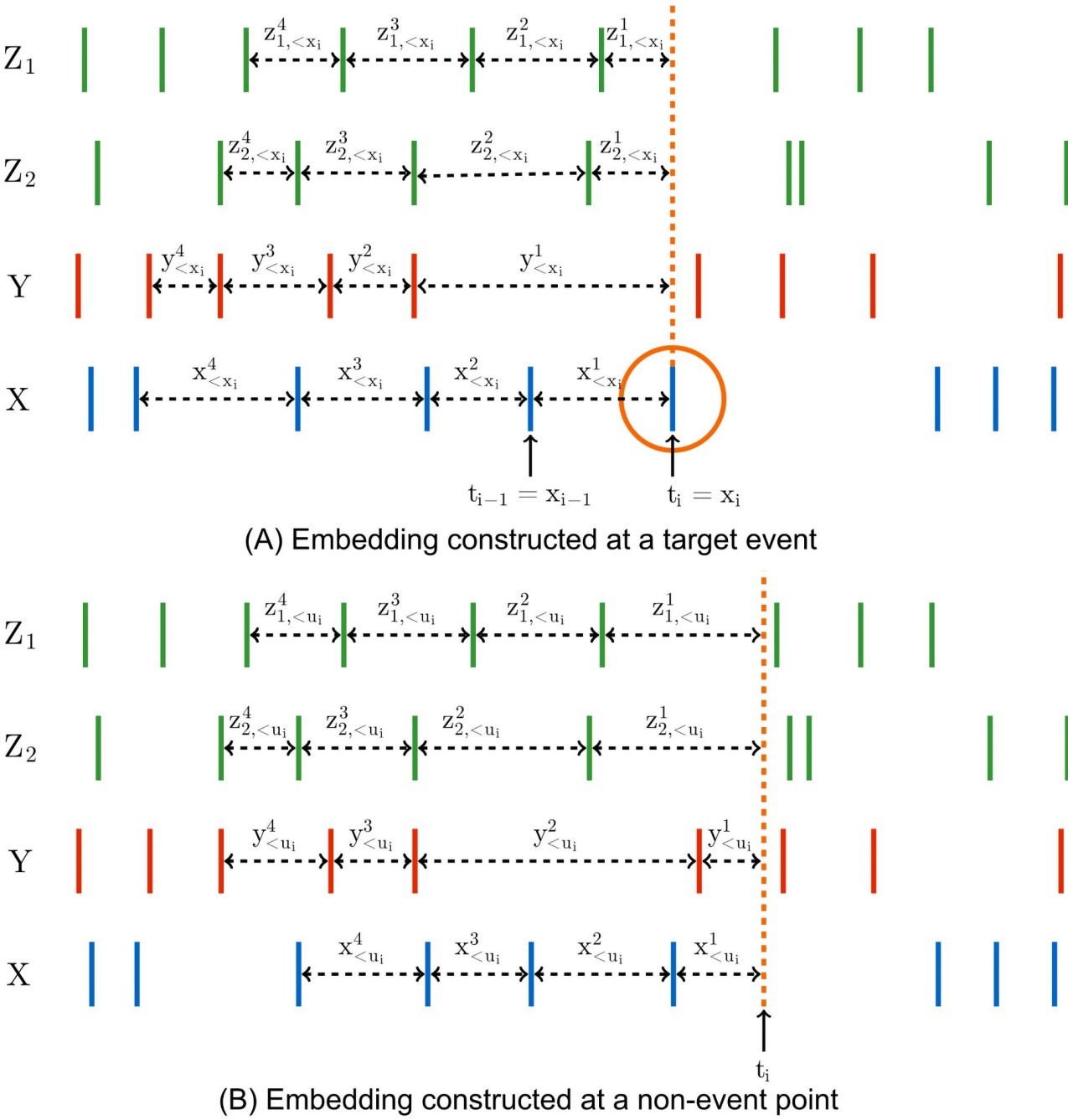

**Fig 10. Examples of history embeddings.** (A) shows an example of a joint embedding constructed at a target event ($\mathbf{j}_{<x_i} \in J_{<X}$). (B) shows an example of a joint embedding constructed at a sample event ($\mathbf{j}_{<u_i} \in J_{<U}$).

have that $J_{<U} = X_{<U} \times Y_{<U} \times \mathbf{Z}_{<U}$. Fig 10 shows diagramatic examples of an embedded sample from $J_{<X}$ as well as one from $J_{<U}$. Notice the distinction that for $J_{<X}$, the $x^1_{<x_i}$ in the embeddings $\mathbf{x}_{<x_i}$ are specifically an interspike interval from the current spike at $t_i = x_i$ back to the previous spike, which is not the case for $J_{<U}$.

The set of samples $C_{<X} \subseteq \mathbb{R}^{l_X + \sum l_{Z_j}}$ for $p_X(\mathbf{x}_{<x}, \boldsymbol{\varkappa}_{<x})$ and $C_{<U} \subseteq \mathbb{R}^{l_X + \sum l_{Z_j}}$ for $p_U(\mathbf{x}_{<x}, \boldsymbol{\varkappa}_{<x})$ are constructed in a similar manner to their associated sets $J_{<X}$ and $J_{<U}$, however, the source embeddings $\mathbf{y}_{<*_i}$ are discarded. We will also have that $C_{<X} = X_{<X} \times \boldsymbol{\mathscr{Z}}_{<X}$ and $C_{<U} = X_{<U} \times \boldsymbol{\mathscr{Z}}_{<U}$.

Note that, as $J_{<X} = X_{<X} \times Y_{<X} \times \boldsymbol{\mathscr{Z}}_{<X}$ and $C_{<X} = X_{<X} \times \boldsymbol{\mathscr{Z}}_{<X}$, these two sets are closely related. Specifically, the $i$-th element of $C_{<X}$ will be identical to the $i$-th element of $J_{<X}$, apart from missing the source component $\mathbf{y}_{<x_i}$. Further, as the same set $U$ is used for both $C_{<U}$ and $J_{<U}$, we will have that the $i$-th element of $C_{<U}$ will be identical to the $i$-th element of $J_{<U}$, apart from missing the source component $\mathbf{y}_{<u_i}$.

**Combining $\hat{H}_*$ estimators for $\hat{T}_{Y \to X | \boldsymbol{\mathscr{Z}}}$.**   With sets of samples and their embedded representation determined as per the previous subsection, we are now ready to estimate each of the four $\hat{H}_*$ terms in Eq (9). Here we consider how to combine the entropy and cross entropy estimators of these terms (Eqs (13) and (15)) into a single estimator.

We could simply estimate each $H^*$ term in Eq (9) using $\hat{H}_{\mathrm{KL}}$ as specified in Eq 13 and $\hat{H}_{p_U, \mathrm{KL}}$ as specified in Eq (15), with the same number $k$ of nearest neighbours in each of the four estimators and at each sample in the set for each estimator. Following the convention introduced in [90] we shall refer to this as a 4KL estimator of transfer entropy (the '4' refers to the 4 $k$NN searches and the 'KL' to Kozachenko-Leonenko):

$$\hat{\mathbf{T}}_{Y \to X | \boldsymbol{\mathscr{Z}}, 4\mathrm{KL}} = \frac{\bar{\lambda}_X}{N_X} \sum_{i=1}^{N_X} \Big\{ \; l_J [ -\ln \epsilon(k, \mathbf{j}_{<x_i}, J_{<X}) + \ln \epsilon(k, \mathbf{j}_{<x_i}, J_{<U}) ]$$
$$+ l_C [ \ln \epsilon(k, \mathbf{c}_{<x_i}, C_{<X}) - \ln \epsilon(k, \mathbf{c}_{<x_i}, C_{<U}) ] \Big\}. \tag{17}$$

Here, $l_J = (l_X + l_Y + \sum_{j=1}^{n_{\boldsymbol{\mathscr{Z}}}} l_{Z_j})$ is the dimension of the joint samples and $l_C = (l_X + \sum_{j=1}^{n_{\boldsymbol{\mathscr{Z}}}} l_{Z_j})$ is the dimension of the conditional-only samples. Note that the $\ln(N_X - 1) - \psi(k)$ terms cancel between the $J_{<X}$ and $C_{<X}$ terms (also for $\ln(N_U) - \psi(k)$ between the $J_{<U}$ and $C_{<U}$ terms), whilst the $\ln c_{d,L}$ terms cancel between $J_{<X}$ and $J_{<U}$ as well as between $C_{<X}$ and $C_{<U}$. It is crucial also to notice that all terms are averaged over $N_X$ samples taken at target events (the cross-entropies which evaluate probability densities using $J_{<U}$ and $C_{<U}$ still evaluate those densities on the samples $\mathbf{j}_{<x_i} \in J_{<X}$ and $\mathbf{c}_{<x_i} \in C_{<X}$, following the definition in Eq (15)), regardless of whether $N_U = N_X$.

It is, however, not only possible to use a different $k$ at every sample, but desirable when the $k$ are chosen judiciously (as detailed below). We shall refer to this as the *generalised $k$NN estimator*:

$$\hat{\mathbf{T}}_{Y \to X | \boldsymbol{\mathscr{Z}}, \mathrm{generalised}} = \frac{\bar{\lambda}_X}{N_X} \sum_{i=1}^{N_X} \Big\{ \psi\big(k_{J_{<X}, i}\big) - \psi\big(k_{J_{<U}, i}\big) - \psi\big(k_{C_{<X}, i}\big) + \psi\big(k_{C_{<U}, i}\big)$$
$$+ l_J [ -\ln \epsilon(k_{J_{<X}, i}, \mathbf{j}_{<x_i}, J_{<X}) + \ln \epsilon(k_{J_{<U}, i}, \mathbf{j}_{<x_i}, J_{<U}) ]$$
$$+ l_C [ \ln \epsilon(k_{C_{<X}, i}, \mathbf{c}_{<x_i}, C_{<X}) - \ln \epsilon(k_{C_{<U}, i}, \mathbf{c}_{<x_i}, C_{<U}) ] \Big\}. \tag{18}$$

Here $k_{A, i}$ is the number of neighbours used for the $i$th sample in set $A$ for the corresponding entropy estimator for that set of samples. By theorems 3 and 4 of [75] this estimator (and, by implication, the 4KL estimator) is consistent. Application of the generalised estimator requires a scheme for choosing the $k_{A, i}$ at each sample. Work on constructing $H^*$ $k$NN estimators for mutual information [53] and KL divergence [75] has found advantages in having certain $H^*$ terms share the same or similar radii, e.g. resulting in lower overall bias due to components of

biases of individual $H^*$ terms cancelling. Given that we have four $H^*$ terms, there are a number of approaches we could take to sharing radii.

Our algorithm, which we refer to as the CT estimator of TE—$\hat{\mathbf{T}}_{Y \to X|\mathbf{Z},\mathrm{CT}}$—is specified in detail in Box 1. Our algorithm applies the approach proposed in [75] (referred to as the 'bias improved' estimator in that work) to each of the Kullback-Leibler divergence terms separately. In broad strokes, whereas Eq (17) uses the same $k$ for each nearest-neighbour search, this estimator uses the same *radius* for each of the two nearest-neighbour searches relating to a given KL divergence term. In practice, this requires first performing searches with a fixed $k$ in order to determine the radius to use. As such, we start with a fixed parameter $k_{\mathrm{global}}$, which will be

---

**Box 1: Algorithm for the CT TE estimator**

**input** : /* The joint history embeddings at the target events and at the sampled points */
$J_{<X} = \{\mathbf{j}_{<x_i}\}_{i=1}^{N_X}; \; J_{<U} = \{\mathbf{j}_{<u_i}\}_{i=1}^{N_U}$
/* The conditioning history embeddings at the target events and at the sampled points */
$C_{<X} = \{\mathbf{c}_{<x_i}\}_{i=1}^{N_X}; \; C_{<U} = \{\mathbf{c}_{<u_i}\}_{i=1}^{N_U}$
/* The average firing rate of the target process */
$\bar{\lambda}_X$
/* The dimension of each element in the conditioning and joint sets */
$l_C; \; l_J$
/* The minimum number of nearest neighbours to consider in any set */
$k_{\mathrm{global}}$

**output:** $\hat{\mathbf{T}}_{Y \to X|\mathcal{Z},\mathrm{CT}}$.

1 $\hat{\mathbf{T}}_{Y \to X|\mathcal{Z},\mathrm{CT}} \leftarrow 0$
2 **for** $i \leftarrow 1$ **to** $N_X$ **do**

 /* Find the radii associated with the history embeddings constructed at events. */
3 $\xi\left(k_{\mathrm{global}}, \mathbf{j}_{<x_i}, J_{<X}\right) \leftarrow \mathtt{findDistanceToKthNearestNeighbour}\left(k_{\mathrm{global}}, \mathbf{j}_{<x_i}, J_{<X}\right)$
4 $\xi\left(k_{\mathrm{global}}, \mathbf{j}_{<x_i}, J_{<U}\right) \leftarrow \mathtt{findDistanceToKthNearestNeighbour}\left(k_{\mathrm{global}}, \mathbf{j}_{<x_i}, J_{<U}\right)$
5 $r_{\mathrm{joint},i} \leftarrow \max\left\{\xi\left(k_{\mathrm{global}}, \mathbf{j}_{<x_i}, J_{<X}\right), \xi\left(k_{\mathrm{global}}, \mathbf{j}_{<x_i}, J_{<U}\right)\right\}$
6 $k_{J_{<X},i} \leftarrow \mathtt{findNumberOfNeighboursInRadius}\left(r_{\mathrm{joint},i}, \mathbf{j}_{<x_i}, J_{<X}\right)$
7 $\epsilon\left(k_{J_{<X},i}, \mathbf{j}_{<x_i}, J_{<X}\right) \leftarrow 2 * \mathtt{findDistanceToKthNearestNeighbour}\left(k_{J_{<X},i}, \mathbf{j}_{<x_i}, J_{<X}\right)$
8 $k_{J_{<U},i} \leftarrow \mathtt{findNumberOfNeighboursInRadius}\left(r_{\mathrm{joint},i}, \mathbf{j}_{<x_i}, J_{<U}\right)$
9 $\epsilon\left(k_{J_{<U},i}, \mathbf{j}_{<x_i}, J_{<U}\right) \leftarrow 2 * \mathtt{findDistanceToKthNearestNeighbour}\left(k_{J_{<U},i}, \mathbf{j}_{<x_i}, J_{<U}\right)$

 /* Find the radii associated with the embeddings constructed at randomly sampled points. */
10 $\xi\left(k_{\mathrm{global}}, \mathbf{c}_{<x_i}, C_{<X}\right) \leftarrow \mathtt{findDistanceToKthNearestNeighbour}\left(k_{\mathrm{global}}, \mathbf{c}_{<x_i}, C_{<X}\right)$
11 $\xi\left(k_{\mathrm{global}}, \mathbf{c}_{<x_i}, C_{<U}\right) \leftarrow \mathtt{findDistanceToKthNearestNeighbour}\left(k_{\mathrm{global}}, \mathbf{c}_{<x_i}, C_{<U}\right)$
12 $r_{\mathrm{conditioning},i} \leftarrow \max\left\{\xi\left(k_{\mathrm{global}}, \mathbf{c}_{<x_i}, C_{<X}\right), \xi\left(k_{\mathrm{global}}, \mathbf{c}_{<x_i}, C_{<U}\right)\right\}$
13 $k_{C_{<X},i} \leftarrow \mathtt{findNumberOfNeighboursInRadius}\left(r_{\mathrm{conditioning},i}, \mathbf{c}_{<x_i}, C_{<X}\right)$
14 $\epsilon\left(k_{C_{<X},i}, \mathbf{c}_{<x_i}, C_{<X}\right) \leftarrow 2 * \mathtt{findDistanceToKthNearestNeighbour}\left(k_{C_{<X},i}, \mathbf{c}_{<x_i}, C_{<X}\right)$
15 $k_{C_{<U},i} \leftarrow \mathtt{findNumberOfNeighboursInRadius}\left(r_{\mathrm{conditioning},i}, \mathbf{c}_{<x_i}, C_{<U}\right)$
16 $\epsilon\left(k_{C_{<U},i}, \mathbf{c}_{<x_i}, C_{<U}\right) \leftarrow 2 * \mathtt{findDistanceToKthNearestNeighbour}\left(k_{C_{<U},i}, \mathbf{c}_{<x_i}, C_{<U}\right)$

 /* We now combine these quantities into the contribution to the TE from the given target event. */
17 $\hat{\mathbf{T}}_{Y \to X|\mathcal{Z},\mathrm{CT}} \leftarrow \hat{\mathbf{T}}_{Y \to X|\mathcal{Z},\mathrm{CT}} + \psi\left(k_{J_{<X},i}\right) - \psi\left(k_{J_{<U},i}\right) - \psi\left(k_{C_{<X},i}\right) + \psi\left(k_{C_{<U},i}\right)$
$\qquad + l_J\left[-\ln\epsilon\left(k_{J_{<X},i}, \mathbf{j}_{<x_i}, J_{<X}\right) + \ln\epsilon\left(k_{J_{<U},i}, \mathbf{j}_{<x_i}, J_{<U}\right)\right]$
$\qquad + l_C\left[\ln\epsilon\left(k_{C_{<X},i}, \mathbf{c}_{<x_i}, C_{<X}\right) - \ln\epsilon\left(k_{C_{<U},i}, \mathbf{c}_{<x_i}, C_{<U}\right)\right]$
18 **end**
19 $\hat{\mathbf{T}}_{Y \to X|\mathcal{Z},\mathrm{CT}} \leftarrow \frac{\bar{\lambda}_X}{N_X}\hat{\mathbf{T}}_{Y \to X|\mathcal{Z},\mathrm{CT}}$

---

the minimum number of nearest neighbours in any search space. For each joint sample at a target event, that is, each $\mathbf{j}_{<x_i}$ in $J_{<X}$, we perform a $k_{\text{global}}$NN search in this same set $J_{<X}$ and record the distance to the $k_{\text{global}}$-th nearest neighbour (line 3 of Box 1). We perform a similar $k_{\text{global}}$NN search for $\mathbf{j}_{<x_i}$ in the set of joint samples independent of target activity $J_{<U}$, again recording the distance to the $k_{\text{global}}$-th nearest neighbour (line 4). We define a search radius as the maximum of these two distances (line 5). We then find the number of points in $J_{<X}$ that fall within this radius of $\mathbf{j}_{<x_i}$ and set $k_{J_{<X}, i}$ as this number (line 6). We also find twice the distance to the $k_{J_{<X}, i}$-th nearest neighbour, which is the term $\epsilon(k_{J_{<X}, i}, \mathbf{j}_{<x_i}, J_{<X})$ in Eq (18) (line 7). Similarly, we find the number of points in $J_{<U}$ that fall within the search radius of $\mathbf{j}_{<x_i}$ and set $k_{J_{<U}, i}$ as this number (line 8). We find twice the distance to the $k_{J_{<U}, i}$-th nearest neighbour, which is the term $\epsilon(k_{J_{<U}, i}, \mathbf{j}_{<x_i}, J_{<U})$ (line 9).

In the majority of cases, only one of these two $\epsilon$ terms will be exactly twice the search radius, and its associated $k_{A,i}$ will equal $k_{\text{global}}$. In such cases, the other $\epsilon$ will be less than twice the search radius and its associated $k_{A,i}$ will be greater than or equal to $k_{\text{global}}$.

The same set of steps is followed for each conditioning history embedding that was constructed at an event in the target process, that is, each $\mathbf{c}_{<x_i}$ in $C_{<X}$, over the sets $C_{<X}$ and $C_{<U}$ (lines 10 through 16 of Box 1).

The values that we have found for $k_{J_{<X}, i}$, $k_{J_{<U}, i}$, $k_{C_{<X}, i}$, $k_{C_{<U}, i}$, $\epsilon(k_{J_{<X}, i}, \mathbf{j}_{<x_i}, J_{<x_i})$, $\epsilon(k_{J_{<U}, i}, \mathbf{j}_{<x_i}, J_{<U})$, $\epsilon(k_{C_{<X}, i}, \mathbf{c}_{<x_i}, C_{<x_i})$ and $\epsilon(k_{C_{<U}, i}, \mathbf{c}_{<x_i}, C_{<U})$ can be plugged into Eq (18) (lines 17 and 19 of Box 1).

**Handling dynamic correlations.**   The derivation of the $k$NN estimators for entropy and cross entropy given above assumes that the points are independent [53]. However, nearby interspike intervals might be autocorrelated (e.g. during bursts), and indeed our method for constructing history embeddings (see Selection and Representation of Sample Histories for Entropy Estimation) will incorporate the same interspike intervals at different positions in consecutive samples. This contradicts the assumption of independence. In order to satisfy the assumption of independence when counting neighbours, conventional neighbour counting estimators can be made to ignore matches within a dynamic or serial correlation exclusion window (a.k.a. Theiler windows [92, 93]).

For our estimator, we maintain a record of the start and end times of each history embedding, providing us with an exclusion window. The start time is recorded as the time of the first event that formed part of an interval which was included in the sample. This event could come from the embedding of any of the processes from which the sample was constructed. The end of the window is the observation point from which the sample is constructed. When performing nearest neighbour and radius searches (lines lines 3, 4, 6, 7, 8, 9, 10, 11, 13, 14, 15 and 16 of Box 1 and line 6 of Box 2), any sample whose exclusion window overlaps with the exclusion window of the original data point around which the search is taking place is ignored. Subtleties concerning dynamic correlation exclusion for surrogate calculations are considered in the next subsection.

## Local permutation method for surrogate generation

A common use of this estimator would be to ascertain whether there is a non-zero conditional information flow between two components of a system. When using TE for directed functional network inference, this is the criteria we use to determine the presence or absence of a connection. Given that we are estimating the TE from finite samples, we require a statistical test in order to determine the significance of the measured TE value. Unfortunately, analytic results do not exist for the sampling distribution of $k$NN estimators of information theoretic

## Box 2: Algorithm for the local permutation method for surrogate generation.

**Input** : /* The joint history embeddings at the target events */

$$J_{<X} = \{\mathbf{j}_{<x_i}\}_{i=1}^{N_X} = \{\mathbf{x}_{<x_i}, \mathbf{y}_{<x_i}, \mathbf{z}_{<x_i}\}_{i=1}^{N_X}$$

/* The joint history embeddings at the sampled points */

$$J_{<U,\mathrm{surr}} = \{\mathbf{j}_{<u_i}\}_{i=1}^{N_{U,\mathrm{surr}}} = \{\mathbf{x}_{<u_i}, \mathbf{y}_{<u_i}, \mathbf{z}_{<u_i}\}_{i=1}^{N_{U,\mathrm{surr}}}$$

$$k_{\mathrm{perm}}$$

**Output** : $J_{<X,\mathrm{surr}}$

/* Set to keep a record of the used indices in the independently sampled embeddings. */

1 $\mathcal{U} \leftarrow \emptyset$ /* Initialise this set to be empty */

2 $J_{<X,\mathrm{surrogate}} \leftarrow \emptyset$ /* Initialise the surrogate embeddings as empty */

3 $I \leftarrow \{i\}_{i=1}^{N_X}$ /* Initialise the indices to iterate over */

/* Shuffle the indices to ensure that different samples are assigned duplicate source componenents each time surrogate sample sets are generated. */

4 $I \leftarrow \mathtt{shuffle}(I)$

5 **for** $i \in I$ **do**

 /* Search for the nearest neighbours in the set of embeddings at sampled points; ignoring the source components. The function findIndicesOfNearestNeighbours(k;a;B) finds the indices of the k nearest neighbours of the point a in the set B. */

6 $\mathcal{N} \leftarrow \mathtt{findIndicesOfNearestNeighbours}\left(k_{\mathrm{perm}}, \{\mathbf{x}_{<x_i}, \mathbf{z}_{<x_i}\}, \{\mathbf{x}_{<u_j}, \mathbf{z}_{<u_j}\}_{j=1}^{N_{U,\mathrm{surr}}}\right)$

 /* Create a set of candidate indices by removing those already used. */

7 $\mathcal{E} \leftarrow \mathcal{N} \setminus (\mathcal{N} \cap \mathcal{U})$

8 **if** $\|\mathcal{E}\| > 0$ **then**

9 $h \leftarrow \mathtt{chooseRandomElement}(\mathcal{E})$

10 **end**

11 **else**

12 $h \leftarrow \mathtt{chooseRandomElement}(\mathcal{N})$

13 **end**

 /* Append the set of surrogate samples with an embedding composed of the original target and conditioning components (at index i); but with the source component swapped for that at index h of the independently sampled embeddings. */

14 $J_{<X,\mathrm{surrogate}} \leftarrow J_{<X,\mathrm{surrogate}} \cup \{\mathbf{x}_{<x_i}, \mathbf{y}_{<u_h}, \mathbf{z}_{<x_i}\}$

 /* Add the index that we chose to the set keeping track of used indices */

15 $\mathcal{U} \leftarrow \mathcal{U} \cup h$

16 **end**

quantities [48]. This necessitates a scheme for generating surrogate samples from which the null distribution can be empirically constructed.

It is instructive to first consider the more general case of testing for non-zero mutual information. As the mutual information between $X$ and $Y$ is zero if and only $X$ and $Y$ are independent, testing for non-zero mutual information is a test for statistical dependence. As such, we are testing against the null hypothesis that $X$ and $Y$ are independent ($X \perp\!\!\!\perp Y$) or, equivalently, that the joint probability distribution of $X$ and $Y$ factorises as $p(X, Y) = p(X)p(Y)$. It is straightforward to construct surrogate pairs $(\check{x}, \check{y})$ that conform to this null hypothesis. We start with the original pairs $(x, y)$ and resample the $y$ values across pairs, commonly by shuffling (in conjunction with handling dynamic correlations, as per Implementation). This shuffling process will maintain the marginal distributions $p(X)$ and $p(Y)$, and the same number of samples, but will destroy any relationship between $X$ and $Y$, yielding the required factorisation for the null

hypothesis. One shuffling process produces one set of surrogate samples; estimates of mutual information on populations of such surrogate sample sets yields a null distribution for the mutual information.

As transfer entropy is a conditional mutual information ($I(X_t ; \mathbf{Y}_{<t} \mid \mathbf{X}_{<t}, \boldsymbol{\mathcal{Z}}_{<t})$), we are testing against the null hypothesis that the current state of the target $X_t$ is conditionally independent of the history of the source $\mathbf{Y}_{<t}$ ($X_t \perp\!\!\!\perp \mathbf{Y}_{<t} \mid \mathbf{X}_{<t}, \boldsymbol{\mathcal{Z}}_{<t}$). That is, the null hypothesis states that the joint distribution factorises as: $p(X_t, \mathbf{Y}_{<t} \mid \mathbf{X}_{<t}, \boldsymbol{\mathcal{Z}}_{<t}) = p(X_t \mid \mathbf{X}_{<t}, \boldsymbol{\mathcal{Z}}_{<t})p(\mathbf{Y}_{<t} \mid \mathbf{X}_{<t}, \boldsymbol{\mathcal{Z}}_{<t})$.

Historically, the generation of surrogates for TE has been done by either shuffling source history embeddings or by shifting the source time series (see discussions in e.g. [4, 94]). These approaches lead to various problems. These problems stem from the fact that they destroy any relationship between the source history ($\mathbf{Y}_{<t}$) and both the target ($\mathbf{X}_{<t}$) and conditioning ($\boldsymbol{\mathcal{Z}}_{<t}$) histories. As such, they are testing against the null hypothesis that the joint distribution factorises as: $p(X_t, \mathbf{Y}_{<t} \mid \mathbf{X}_{<t}, \boldsymbol{\mathcal{Z}}_{<t}) = p(X_t \mid \mathbf{X}_{<t}, \boldsymbol{\mathcal{Z}}_{<t})p(\mathbf{Y}_{<t})$ [48]. The problems associated with this factorisation become particularly pronounced when we are considering a system whereby the conditioning processes $\boldsymbol{\mathcal{Z}}_{<t}$ drive both the current state of the target $X_t$ as well as the history of the source $\mathbf{Y}_{<t}$. This can lead to $\mathbf{Y}_{<t}$ being highly correlated with $X_t$, but conditionally independent. This is the classic case of a "spurious correlation" between $\mathbf{Y}_{<t}$ and $X_t$ being mediated through the "confounding variable" $\boldsymbol{\mathcal{Z}}_{<t}$. If, in such a case, we use time shifted or shuffled source surrogates to test for the significance of the TE, we will be comparing the TE measured when $X_t$ and $\mathbf{Y}_{<t}$ are highly correlated (albeit potentially conditionally independent) with surrogates where they are independent. This subtle difference in the formulation of the null may result in a high false positive rate in a test for conditional independence. An analysis of such a system is presented in the third subsection of Results. Alternately, if we can generate surrogates where the joint probability distribution factorises correctly and the relationship between $\mathbf{Y}_{<t}$ and the histories $\mathbf{X}_{<t}$ and $\boldsymbol{\mathcal{Z}}_{<t}$ is maintained, then $\mathbf{Y}_{<t}$ will maintain much of its correlation with $X_t$ through the mediating variables $\boldsymbol{\mathcal{Z}}_{<t}$ and $\mathbf{X}_{<t}$. We would anticipate conditional independence tests using surrogates generated under this properly formed null to have a false positive rate closer to what we expect.

Generating surrogates for testing for conditional dependence is relatively straightforward in the case of discrete-valued conditioning variables. If we are testing for dependence between $X$ and $Y$ given $Z$, then, for each unique value of $Z$, we can shuffle the associated values of $Y$. This maintains the distributions $p(X|Z)$ and $p(Y|Z)$ whilst, for any given value of $Z$, the relationship between the associated $X$ and $Y$ values is destroyed.

The problem is more challenging when $Z$ can take on continuous values. However, recent work by Runge [48], demonstrated the efficacy of a local permutation technique. In this approach, to generate one surrogate sample set, we separately generate a surrogate sample $(x, \check{y}, z)$ for each sample $(x, y, z)$ in the original set. We find the $k_{\text{perm}}$ nearest neighbours of $z$ in $Z$: one of these neighbours, $z'$, is chosen at random, and $y$ is swapped with the associated $y'$ to produce the surrogate sample $(x, y', z)$. In order to reduce the occurrence of duplicate $y$ values, a set $\mathcal{U}$ of used indices is maintained. After finding the $k_{\text{perm}}$ nearest neighbours, those that have already been used are removed from the candidate set. If this results in an empty candidate set, one of the original $k_{\text{perm}}$ candidates are chosen at random. Otherwise, this choice is made from the reduced set. As before, a surrogate conditional mutual information is estimated for every surrogate sample set, and a population of such surrogate estimates provides the null distribution.

This approach needs to be adapted slightly in order to be applied to our particular case, because we have implictly removed the target variable (whether or not the target is spiking)

from our samples via the novel Bayesian inversion. We can rewrite Eq (8) as:

$$\dot{\mathbf{T}}_{Y \to X | \boldsymbol{\mathcal{Z}}} = \bar{\lambda}_X \mathbb{E}_X \left[ \ln \frac{p_X(\mathbf{x}_{<x}, \mathbf{y}_{<x}, \boldsymbol{\mathcal{z}}_{<x})}{p_X(\mathbf{x}_{<x}, \boldsymbol{\mathcal{z}}_{<x}) p_U(\mathbf{y}_{<x} \mid \mathbf{x}_{<x}, \boldsymbol{\mathcal{z}}_{<x})} \right]. \tag{19}$$

This makes it clear that we are testing whether the following factorisation holds:

$$p_X(\mathbf{x}_{<x}, \mathbf{y}_{<x}, \boldsymbol{\mathcal{z}}_{<x}) = p_X(\mathbf{x}_{<x}, \boldsymbol{\mathcal{z}}_{<x}) p_U(\mathbf{y}_{<x} \mid \mathbf{x}_{<x}, \boldsymbol{\mathcal{z}}_{<x}) \tag{20}$$

(recall the difference between probability densities at target events $p_X$ and those not conditioned at target events $p_U$). In order to create surrogates $J_{<X,\text{surr}}$ that conform to this null distribution, we resample a new set from our original data in a way that maintains the relationship between the source histories and the histories of the target and conditioning processes, but decouples (only) the source histories from target events. (As above, simply shuffling the source histories across $J_{<X}$ or shifting the source events does not properly maintain the relationship of the source to the target and conditioning histories). The procedure to achieve this is detailed in Box 2. We start with the samples at target events $J_{<X} = \left\{ \mathbf{x}_{<x_i}, \mathbf{y}_{<x_i}, \boldsymbol{\mathcal{z}}_{<x_i} \right\}_{i=1}^{N_X}$ and resample the source components $\mathbf{y}_{<x_i}$ as follows. We first construct a new set $J_{<U,\text{surr}} = \left\{ \mathbf{x}_{<u_i}, \mathbf{y}_{<u_i}, \boldsymbol{\mathcal{z}}_{<u_i} \right\}_{i=1}^{N_{U,\text{surr}}}$ from the set $U_{\text{surr}}$ of $N_{U,\text{surr}}$ points sampled independently of events in the target. This set is constructed in the same manner as $J_{<U}$, although we might choose to change the number of sample points ($N_{U,\text{surr}} \neq N_U$) at which the embeddings are constructed, or whether the points are placed randomly or at fixed intervals. For each original sample $\mathbf{j}_{<x_i}$ from $J_{<X}$, we then find the nearest neighbours $\left\{ \mathbf{x}_{<u_i}, \boldsymbol{\mathcal{z}}_{<u_i} \right\}_{i=1}^{k_{\text{perm}}}$ of $\left\{ \mathbf{x}_{<x_i}, \boldsymbol{\mathcal{z}}_{<x_i} \right\}$ in $J_{<U,\text{surr}}$ (line 9 of Box 12), select $\mathbf{y}_{<u_j}$ randomly from amongst the $k_{\text{perm}}$ nearest neighbours (line 6 or Box 2), and add a sample $\left\{ \mathbf{x}_{<x_i}, \mathbf{y}_{<u_j}, \boldsymbol{\mathcal{z}}_{<x_i} \right\}$ to $J_{<X,\text{surr}}$ (line 14). The construction of such a sample is also displayed in Fig 11. Similar to Runge [48], we also keep a record of used indices in order

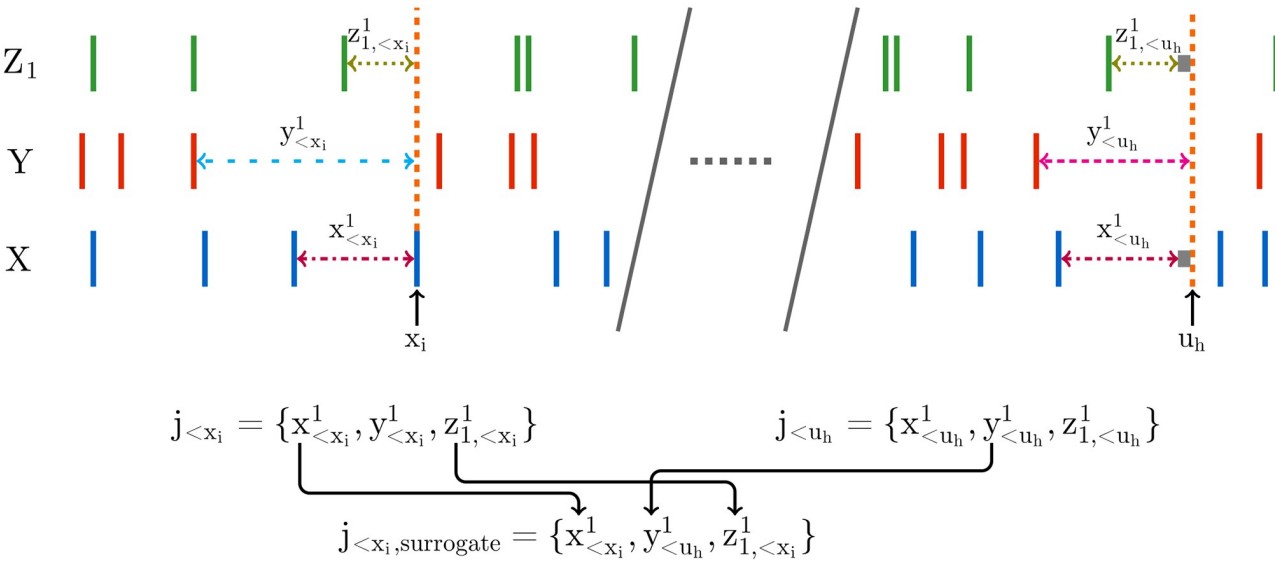

**Fig 11. Diagrammatic representation of the local permutation surrogate generation scheme.** For our chosen sample $\mathbf{j}_{<x_i}$ we find a $\mathbf{j}_{<u_h} \in J_{<U,\text{surr}}$ where we have that the $\mathbf{x}_{<x_i}$ component of $\mathbf{j}_{<x_i}$ is similar to the $\mathbf{x}_{<u_h}$ component of $\mathbf{j}_{<u_h}$ and $\boldsymbol{\mathcal{z}}_{<x_i}$ component of $\mathbf{j}_{<x_i}$ is similar to the $\boldsymbol{\mathcal{z}}_{<u_h}$ component of $\mathbf{j}_{<u_h}$. We then form a single surrogate sample by combining the $\mathbf{x}_{<x_i}$ and $\boldsymbol{\mathcal{z}}_{<x_i}$ components of $\mathbf{j}_{<x_i}$ with the $\mathbf{y}_{<u_h}$ component of $\mathbf{j}_{<u_h}$. Corresponding colours of the dotted interval lines indicates corresponding length. The grey boxes indicate a small delta.

to reduce the incidence of duplicate $\mathbf{y}_{<u_j}$ (line 15). For each redrawn surrogate sample set $J_{<X, \text{ surr}}$ a surrogate conditional mutual information is estimated (utilising the same $J_{<U}$ selected independently of the target events as was used for the original TE estimate) following the algorithm outlined earlier; the population of such surrogate estimates provides the null distribution as before.

The $p$ values are calculated by constructing $N_{\text{surrogates}}$ surrogates by the algorithm just described. The TE is estimated on these surrogates and compared to the TE estimated on the original embeddings. The $p$ value is then the number of estimates on surrogate embeddings which were larger than the estimate on the original data divided by the total number of surrogates.

Finally, we note an additional subtlety for dynamic correlation exclusion for the surrogate calculations. Samples in the surrogate calculations will have had their history components originating from two different time windows. One will be from the construction of the original sample and the other from the sample with which the source component was swapped. A record is kept of both these exclusion windows and, during neighbour searches, points are excluded if their exclusion windows intersect either of the exclusion windows of the surrogate history embedding.

## Implementation

The algorithms shown in Boxes 2 and 1 as well as all experiments were implemented in the Julia language. The implementation of the algorithms is freely available at the following repository: github.com/dpshorten/CoTETE.jl. Scripts to run the experiments in the paper can be found here: github.com/dpshorten/CoTETE_experiments. Implementations of $k$NN information-theoretic estimators have commonly made use of KD-trees to speed up the nearest neighbour searches [94]. A popular Julia nearest neighbours library (NearestNeighbors.jl, available from github.com/KristofferC/NearestNeighbors.jl) was modified such that checks for dynamic exclusion windows (see Handling Dynamic Correlations) were performed during the KD-tree searches when considering adding neighbours to the candidate set.

## Assumptions used to conclude conditional independence or dependence

We summarise here the conditions and assumptions that allow us to draw conclusions about conditional independence relationships from the structure in a model. Although these relationships are obvious in some of our examples (see Results), they are less so in others. If we consider, for now, the discrete-time case, then for a sufficiently small $\Delta t$ there will be no instantaneous effects. This implies that the causal relationships in these models can be represented by a Directed Acyclic Graph (DAG); specifically a Dynamic Bayesian Network with multiple time slices and connections only going forward in time (see [95]). In order to conclude that connected nodes will be statistically dependent we need to use the contraposition of the faithfulness assumption [55, 56, 57]. This assumption states that, if two nodes are conditionally independent, given some conditioning set $S$, then they are $d$-separated [55] given the same set. This in turn implies that if there exists some set of conditioning processes by which a node is conditionally independent of another, then there is no direct causal link between these nodes. It is worth asking how reasonable the faithfulness assumption is. After all, particularly for the case of deterministic dynamics, it is easy to construct examples whereby the present state of each of a pair of processes is determined by the history of the other process, but where the present state of each process is conditionally independent of the history of the other [96, 97] (e.g. one can have zero TE when a real causal connection exists, for instance, the system $x_t = y_{t-1}$, $y_t = x_{t-1}$, $x_1 = 0$ and $y_1 = 1$). Such examples violate

faithfulness. However, determinism is not a realistic assumption for biological systems or their models. Moreover, it can be shown that almost all discrete probability distributions (such as those of spike trains) satisfy faithfulness. Indeed, the set of discrete probability distributions that violate this assumption has measure zero [98]. In order to determine that the present state of a process is independent of an unconnected source, when conditioning on its direct causal parents, we need to assume sufficiency and the causal Markov condition [55, 56, 57]. Sufficiency assumes that we have observed all relevant variables (which is easy to meet if we are defining the model). The causal Markov condition states that $d$-separation implies conditional independence. Conditioning on all the direct causal parents of a variable provides us $d$-separation. In summary then, under these conditions the directed structural connections designed in our models are expected to have a one-to-one correspondence with directed conditional dependence (or independence, in their absence), when appropriately conditioned on other nodes. Correctly differentiating conditional dependence and independence then, in alignment with the underlying structural connections in these models, provides an important validation of the correctness of the estimators.

## Specification of leaky-integrate-and-fire model

What follows is a specification of the Leaky-Integrate-and-Fire (LIF) model which we used in the Results subsection Scaling of Conditional Independence Testing in Higher Dimensions. The membrane potential evolves according to:

$$\tau \frac{dV}{dt} = V_0 - V. \tag{21}$$

When $V$ crosses the threshold $V_{\text{threshold}}$, the timestamp of crossing is recorded as a spike. $V$ is then set to $V_{\text{reset}}$ and the evolution of the membrane potential is subsequently paused for the duration of the hard refractory period. In the case of excitatory connections, when a presynaptic spike occurs, $V$ is instantaneously increased by the connection strength of the synapse (specified in millivolts) at the delay specified by the connections delay parameter. Inhibitory connects behave in the same manner, but lead to a decrease in $V$. We use the initial condition $V(t = 0) = V_0$.

## Supporting information

**S1 Fig. Longer embeddings on homogeneous.** The results of an identical experimental setup to those displayed in Fig 2, but with history embedding lengths of $l_X = l_Y = 3$.
(TIFF)

**S2 Fig. Different embeddings on discrete homogeneous.** The results of an identical experimental setup to those displayed in Fig 3, but where the history embedding lengths ($l$ and $m$) were set to cover the distance of an average interspike interval. Specifically, these lengths were 1, 2, 5 and 10, corresponding to the $\Delta t$ values of 1.0, 0.5, 0.2 and 0.1.
(TIFF)

**S3 Fig. Different embeddings on continuous coupled.** The results of an identical experimental setup to those displayed in Fig 4B, but where the history embedding length of the source is set to $l_Y = 3$.
(TIFF)

**S4 Fig. Conditional independence scaling at constant rate.** The results of an identical experimental setup to those displayed in Fig 8, but with a constant rate of 20 Hz in all the stimuli. This removes the correlation between the unconnected source and the firing of the target. The

top row shows results of the continuous-time approach, the bottom shows results of the discrete-time approach.
(TIFF)

**S5 Fig. Independence test with no conditioning.** The results of an identical experimental setup to those displayed in Fig 8, but where the background processes are not included in the conditioning set (the conditioning set is left empty). This represents the nature of the inference task at the early stage of a greedy network inference algorithm being applied to a node. We see that the continuous-time estimator performs well on inhibitory connections in this case. Due to the change in dimension, different source and target embedding lengths ($l$ and $m$) as well as bin widths $\Delta t$ were used for the discrete-time estimator. These were set at $l = m = 12$ and $\Delta t = 2$ms. The top row shows results of the continuous-time approach, the bottom shows results of the discrete-time approach.
(TIFF)

**S6 Fig. Conditional independence testing with the discrete-time estimator and permutation-based surrogates.** The results of identical experimental setups to those displayed in the bottom rows of Fig 8, S4 and S5 Figs. As the bottom rows of all of these figures show the results of the discrete-time estimator, the plots in this figure similarly all display the results of runs of the discrete-time estimator. However, where the other plots make use of the source time-shift method for surrogate generation (as is traditionally used in conjunction with TE estimators), these plots make use of a standard conditional-permutation-based surrogate generation scheme for categorical variables [64]. The top row of this figure corresponds to the bottom row of Fig 8, the middle row corresponds to the middle row of S4 Fig and the bottom row corresponds to the bottom row of S5 Fig.
(TIFF)

**S7 Fig. Pyloric STG continuous different embedding lengths.** The results of an identical experimental setup to those displayed in Fig 9C, but where different embeddings lengths ($l_X$, $l_Y$ and $l_{Z_1}$) are used. The left plot shows $l_X = l_Y = l_{Z_1} = 2$ and the right plot shows $l_X = l_Y = l_{Z_1} = 4$.
(TIFF)

**S8 Fig. Pyloric STG continuous different dataset sizes.** The results of an identical experimental setup to those displayed in Fig 9C, but where different numbers of target spikes $N_X$ are used. The left plot shows $N_X = 1 \times 4$ and the right plot shows $N_X = 3.5 \times 4$.
(TIFF)

**S1 Text. Description of the biophysical neural network model insipred by the Pyloric STG.**
(PDF)

## Acknowledgments

We would like to thank Mike Li for performing preliminary benchmarking of the performance of the discrete-time estimator on point processes.

## Author Contributions

**Conceptualization:** David P. Shorten, Richard E. Spinney, Joseph T. Lizier.

**Funding acquisition:** Joseph T. Lizier.

**Methodology:** David P. Shorten, Richard E. Spinney, Joseph T. Lizier.

**Software:** David P. Shorten.

**Supervision:** Richard E. Spinney, Joseph T. Lizier.

**Writing – original draft:** David P. Shorten.

**Writing – review & editing:** David P. Shorten, Richard E. Spinney, Joseph T. Lizier.

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
