## [Decision Letter · Decision Letter 0]

19 Jul 2020

Dear Mr Shorten,

Thank you very much for submitting your manuscript "Estimating Transfer Entropy in Continuous Time Between Neural Spike Trains or Other Event-Based Data" for consideration at PLOS Computational Biology.

As with all papers reviewed by the journal, your manuscript was reviewed by members of the editorial board and by several independent reviewers. In light of the reviews (below this email), we would like to invite the resubmission of a significantly-revised version that takes into account the reviewers' comments. Also randomising the source and not the target is probably a typo (at least I hope so ;)), you might want to fix it. 

We cannot make any decision about publication until we have seen the revised manuscript and your response to the reviewers' comments. Your revised manuscript is also likely to be sent to reviewers for further evaluation.

Sincerely,

Daniele Marinazzo

Deputy Editor

PLOS Computational Biology

Daniele Marinazzo

Deputy Editor

PLOS Computational Biology

Reviewer's Responses to Questions

**Comments to the Authors: **

Reviewer #1: This paper introduces a method and an empirical approach for estimating transfer entropy for event-based data. It avoids the time-descritization by operating on the probability densities of the history embeddings of the variables, rather than making a direct estimation of spike rates. Authors also provide an adaptation of a recently proposed surrogate generation scheme in order to have a complete statistical test to provide statistical strength of the conditional independence under consideration. 

Major comments:

Overall the paper is well written. However, special care should be taken in the causal claims and statements that authors make. How will the proposed method perform in the case that Xt-1 -> Yt and Yt-1 -> Xt and Xt-s -\\-> Xt and Yt-s -\\-> Yt, for all s>0, t>0 ? According to Janzing et al. 2013 (see section 5.2 in Quantifying causal influences, The Annals of Statistics 41 (5), 2324-2358.), transfer entropy will always converge to 0 in this particular example although there is causal influence between X and Y. That means that there are cases in which TE will not give correct information. Therefore, authors should mild the sentences were they connect TE estimation to causal statements and they should also discuss that their TE estimator and surrogate scheme assumes acyclic summary graphs and causal sufficiency.

Line 46: Authors should define the meaning of "active" causal connection between X and Y. Do you imply a connection that does not include any hidden variable? If that is what the authors want to say they need to mention they assume causal sufficiency.

Line 49: Authors should particularly present the assumptions they imply here. Do they mean the necessary assumption for relating conditional independence tests and d-separation statements in causal graphs? ( "Causal Faithfulness" and "Causal Markov Condition").

Line 179: Authors claim that non-zero TE values imply a directed relationship. That is not true if Causal Sufficiency (all common causes are observed) is not assumed. In presence of hidden confounders Y <- H -> X the TE for Y and X will be different from zero, as H cannot be observed. Nevertheless this is not a directed relationship between Y and X. Authors need to write that this is true only under causal sufficiency.

Figure 2: The figure writes Ns/Nx while the caption and in the text writes Nu/Nx. Please be consistent with your notation across the main text and the figures. 

Line 208: The authors write "the averages are taken across 1000, 100, 100 and 100 tested process pairs respectively." Why the first one is averaged over 1000 and the other three over 100?

Line 262: Causal statements should be made with cautiousness. Authors need to specify the assumptions they imply. Causal faithfulness and Causal Markov Condition (if these are the implied ones) are not weak assumptions. In addition causal sufficiency which is necessary for granger causality is a very strict and hard to meet assumption. Its violation renders TE inadequate for causal discovery.

Line 491: Regarding the successful application of the method on the pyloric network of the crustacean stomatogastric ganglion: I have some questions. First of all, the referenced paper that used Granger Causality, used a linear implementation. Does the preprocessing of the data differ between the two papers? Did the authors compared directly on their data with GC? If so, did they try both linear and non linear implementations of GC? Since both GC and the authors' method suffers from hidden confounding both would be expected to fail in a real scenario with hidden confounders. Were all the necessary information observed in the presented experiment? If yes, then it should be stressed accordingly because this is an ideal scenario. Could the authors elaborate on their intuition why their method worked successfully compared to Granger? Is it because their method is specifically designed for event-based data? Could the linear implementation be the reason of the many false positives of Granger method? 

Also, it would be useful to provide TE estimation results compared with Runge's (Runge et al. 2018 Escaping the Curse of Dimensionality in Estimating Multivariate Transfer Entropy) and Gao's (Gao et al.2016 A Transfer Entropy Method to Quantify Causality in Stochastic Nonlinear Systems) methods. I understand the fundamental extension of the proposed method compared to the other methods, but it would be interesting to see how poorly the other methods perform if that is the case. Finally, how is the proposed method related with the "Instantaneous Transfer Entropy for the Study of Cardio-Respiratory Dynamics" of Valenz et al. and the "Optimization of relative parameters in transfer entropy estimation and application to corticomuscular coupling in humans" of Zhou et al?

Minor: 

This is a personal preference but in general X in causal terminology is used for potential cause/source and Y for target variables. Nevertheless, authors are consistent with the way they define X and Y, so this is not a thing that I insist on changing.

Line 100: "of the of the" repeated twice

Line 856: "through the mediating variable Z<t", it is better to write through the common cause variable Z<t. Mediating is mostly used for chains Y->Z->X.

Line 893: "then, we then find" -> delete one "then"

Line 1118 reads weird

Reviewer #2: Review for "Estimating Transfer Entropy in Continuous Time Between Neural Spike Trains for Other Event-Based Data"

In this manuscript, the authors present a method for estimating transfer entropy from discrete event time series data. They develop the method using k-nearest neighbors estimates of relevant entropic terms. The method is shown to produce better estimates of transfer entropy as compared to other typical methods that discretize the data using time bins, making use of a series of constructed test examples. (One test example is a pre-existing model of bursty neural firing.) The authors also construct a novel method for creating surrogate shuffled datasets to be used for significance tests, identifying and curing a problem that incorrectly breaks conditional correlations when using naive time shuffling approaches.

Overall, the manuscript is well written and convincingly makes its case that the proposed method is better than naive binning procedures for estimating transfer entropy in these cases. The method would be immediately applicable to common spiking datasets in neuroscience.

The development is comprehensive mathematically, though I wonder if a bit more discussion about biological applications would be useful for this audience. For instance, when does the proposed method fail? Does the amount of data required for resolving connectivity depend on the amount of noise in the system? Do you expect the method to successfully scale up to cases with, for instance, hundreds or thousands of neurons? I realize these may only be addressable in future work, but any preliminary thoughts here would be useful.

Other comments:

-- Eq (1) seems to be missing an average over possible states and histories? (Or if not, a brief comment would be nice to orient the reader.)

-- The notation of Eq (3) and (4) was confusing to me before reading the Methods. Specifically, it is not immediately clear that \\lambda depends on time. (Should t be t_i?)

-- Transfer entropy rates are quoted as nats/second (e.g. in Fig 2), but it was not immediately clear to me where the timescale of seconds came from.

-- Should N_S/N_X in Fig 2 be N_U/N_X?

-- Text in figures is extremely small and hard to read.

-- line 368: "a p value of zero to every single example..." This isn't evident from Fig 7---it looks like there are some nonzero p values for sigma_D = 5e-2?

Typo:

-- line 100: "of the"

Reviewer #3: See attachment

Reviewer #4: The review comment will be attached as a document.

**Have all data underlying the figures and results presented in the manuscript been provided?**

Reviewer #1: Yes

Reviewer #2: **No: **The authors have already made the code available on github, but it's not clear that numerical data for figures is provided in spreadsheet form.

Reviewer #3: **No: **The authors have provided a repository with scripts to generate the data

Reviewer #4: Yes

PLOS authors have the option to publish the peer review history of their article (what does this mean?). If published, this will include your full peer review and any attached files.

Reviewer #1: No

Reviewer #2: No

Reviewer #3: **Yes: **Dimitra Maoutsa

Reviewer #4: No
---

## [Decision Letter · Decision Letter 1]

7 Nov 2020

Dear Mr Shorten,

Thank you very much for submitting your manuscript "Estimating Transfer Entropy in Continuous Time Between Neural Spike Trains or Other Event-Based Data" for consideration at PLOS Computational Biology.

The resubmission, where many issues have been addressed, has been generally appreciated.

Nonetheless a few critical points, related to both the foundation of the method, and the presentation of the results, still remain, and we would insist for you to make a substantial effort in this direction.

In light of the reviews (below this email), we would like to invite the resubmission of a significantly-revised version that takes into account the reviewers' comments.

We cannot make any decision about publication until we have seen the revised manuscript and your response to the reviewers' comments. Your revised manuscript is also likely to be sent to reviewers for further evaluation.

Sincerely,

Daniele Marinazzo

Deputy Editor

PLOS Computational Biology

Daniele Marinazzo

Deputy Editor

PLOS Computational Biology

Reviewer's Responses to Questions

**Comments to the Authors:**

Reviewer #1: Comments on Authors' responses and revised manuscript:

I appreciate the fact that the authors added new experiments and their effort to distinguish their method from causal inference methods. However, I believe that there are still some remaining points that this is not completely clear to the reader. I believe that the way the authors presented their method and the way they used causal graph discovery on their simulated neuronal example to validate their method, caused confusion to almost all the reviewers - including myself-, shadowing the importance of the method itself. The contribution of the paper is - and as such should be presented as - a TE estimator that avoids time descritazion, combined with a surrogate scheme. However, even after the revision, to me it seems that there are still misunderstandings for the reader, with claims of the authors about detection of directed dependencies. The method, as it is, is very important and novel. This is what the authors should state. Any further attempts to connect it with directionality of dependencies, only leads to misinterpretations.

My remaining problem after the revision is that authors mix the assumptions needed for TE estimation, the assumptions needed for a conditional independence test and those needed for causal discovery, in their statements.

Overall I do not agree with the fact that the authors advertise this method as a conditional independence test, without specification of the acyclicity assumption that is required. After the revision, the authors added section D where they discuss assumptions for causal discovery on their neuron experiment, using their method. Nevertheless these assumptions are to make causal inference. However, authors promote their method as a conditional independence test without naming the assumptions under which this test holds. Authors mix in their statements the assumptions needed for this method to be an independence test and the assumptions needed for causal inference. Therefore, for me the following points should become extremely clear in order to recommend the paper for acceptance:

1. (self-contradictory points in authors paper/response) In author's response 1.1: Authors write "Along these lines, we do not make the claim that non-zero TE implies a causal connection in general.", which is not in agreement with what they write in their paper line 216 "The comparison of the estimates to a population of surrogates produces p-values for the statistical significance of the TE values being distinct from zero and implying a directed statistical dependence."

In their paper, the authors do claim detection of directed dependencies when TE is not 0 (line 216). This sentence is both in contradiction to their response, and does not stand alone, for the reasons I explain in point 1.2 below. My point in my original comment was that the opposite does not hold: TE=0 does not mean lack of dependency in general, and this is what I asked authors to add in their revision, but they did not. As I explain below, acyclicity between the time series is required for the opposite to hold.

2. (directed dependencies) On authors answer to Response 1.4: Authors write "This sentence does not imply causality. Causal relationships are not the only type of relationship. This sentence is simply reflecting the fact that TE implies that the present of the target is statistically dependent on the history of the source"

Although causal relationships may not be the only type of relationships, the *directed* relationships *are* causal relationships. I do not agree with what the authors write and with the way they update their text. Although their second sentence in the above quoted answer is correct, the updated text does not reflect their statement, as it still writes about directionality of dependencies. " TE values being distinct from zero and implying a directed statistical dependence". The moment the authors talk about directionality, it is not possible to *not* talk about causal relationship. Therefore, either they should completely drop their claims about *directed* relationships and actually write the correct sentence they gave as an answer (aka "TE implies that the present of the target is statistically dependent on the history of the source"), or they should explicitly state that causal sufficiency is assumed in the case that they want to talk about directed relationships. Otherwise the counterexample I gave the authors in my original review will render TE inadequate for any such statements. Same for the revised sentence in point 2 updated text 1.1.

3. (ground truth on simulated experiment) The authors use one of their own methods to calculate the ground truth TE on the experiments with the Poisson processes. As reviewer 4 mentioned, someone, could argue about the choice of this ground truth. I would like to see a longer discussion about the selection of this ground truth. Even if we leave this aside, up to this point I could have no objection. However, moving to the simulation neural example, the authors change their ground truth estimation and use the underlying known structure to evaluate their method. Now, this is causing a lot of confusion, because on one hand authors claim this is not a causal inference method, and on the other hand use it in such a way to validate their estimator. This changes the metric that the authors use to validate their method.

Please add the word "simulated" in the title of section E. As it is presented now, it can lead to overestimation of the ability of the proposed TE method to work as a causal discovery tool in real, heavily latent-confounded noisy datasets, as the neuronal activity.

4. (conditional independence testing) Re the comment of the authors that all conditional independence tests suffer from dimensionality or small sample size (line 753).

It is different to talk about statistical strength of a test (in that case indeed dimensionality and small sample size play an important role about the rejection of conditional independencies), and to talk about a TE estimation used to quantify a dependency. Here TE is used to quantify the dependency and the surrogate scheme to detect the p-value. The proposed method as a dependency estimator does not work properly if acyclicity is not assumed. Therefore the method can be considered only a *sufficient* but not a necessary conditional independence oracle, if the necessary assumptions are not mentioned. If the proposed TE method gives non-zero value, then they conclude dependency. However, the other direction does not hold. If the TE is zero then this does not necessarily mean lack of dependency. The counterexample I explained in my review is the reason why. A correlation permutation test, given a sufficient sample size, would find a dependency between X and Y if Xt−1→Yt and Yt−1→Xt and Xts→Xt and Yts →Yt, for all s>0,t>0. However the proposed conditional independence test, even with sufficient sample size, will find no dependency between X and Y because TE=0 in this example. Therefore the size of the conditioning set or the sample size is not the only problem of the proposed TE-based conditional independence test, as TE will always be zero in this counter example. Therefore, all the following points throughout the paper should be changed accordingly including the necessary assumptions, so that the claim of conditional independence hold:

- "Our approach is shown to be capable of detecting conditional independence or otherwise even in the presence of strong pairwise time-directed correlations" in the abstract

- line 164 "This adapted scheme produces surrogates which conform to the correct null hypothesis of conditional independence of the present of the target and the source history, given..."

- line 302: remove "weak"

- line 304 " More importantly, TE can be used to test for such conditional independence "

- line 311 " In this subsection, we demonstrate that the combination of the presented estimator and surrogate generation scheme is particularly adept at identifying conditional independence in the face of strong pairwise correlations on a synthetic example. "  As I said before, this is a misleading sentence. The reason why the proposed method works here as a CI test is because this is a simple synthetic example that takes into account very serious assumptions, which otherwise would be violated in real datasets.

- line 424 "As argued in sections I and IV B this is equivalent to a test for conditional independence."  The sentence alone is not true. It requires the above acyclycity assumption.

- line 486 "This is likely due to this scheme’s poor ability to identify conditional independence in the presence of pairwise correlations, as we have already seen in Sec. II C."  Both the old and the new method suffer from the same problem and are biased into detecting more independencies if the acyclicity assumption is violated.

To conclude, I believe that the TE estimator the authors propose is a novel, important contribution that overcomes time-districtization. I am not convinced about its use as a conditional independence test, though. Therefore, I would recommend acceptance of the paper, only given that the above points became clear, the authors clearly stated the necessary assumptions and made milder the mis-formulated statements as suggested above.

Reviewer #2: The authors have sufficiently addressed my comments and I support the publication of the manuscript.

Reviewer #3: See attached file

Reviewer #4: Refer the attached file.

**Have all data underlying the figures and results presented in the manuscript been provided?**

Reviewer #1: Yes

Reviewer #2: None

Reviewer #3: Yes

Reviewer #4: Yes

PLOS authors have the option to publish the peer review history of their article (what does this mean?). If published, this will include your full peer review and any attached files.

Reviewer #1: No

Reviewer #2: No

Reviewer #3: **Yes: **Dimitra Maoutsa

Reviewer #4: No
---

## [Decision Letter · Decision Letter 2]

19 Feb 2021

Dear Mr Shorten,

We are pleased to inform you that your manuscript 'Estimating Transfer Entropy in Continuous Time Between Neural Spike Trains or Other Event-Based Data' has been provisionally accepted for publication in PLOS Computational Biology.

Best regards,

Daniele Marinazzo

Deputy Editor

PLOS Computational Biology

Daniele Marinazzo

Deputy Editor

PLOS Computational Biology

Reviewer's Responses to Questions

**Comments to the Authors:**

Reviewer #1: The clarifications the authors added on the paper after my comments on their 1st revision has covered my concerns, therefore I recommend for acceptance.

**Have all data underlying the figures and results presented in the manuscript been provided?**

Reviewer #1: Yes

PLOS authors have the option to publish the peer review history of their article (what does this mean?). If published, this will include your full peer review and any attached files.

Reviewer #1: No

---

## [Editor Report · Acceptance letter]

23 Mar 2021

PCOMPBIOL-D-20-00997R2 

Estimating Transfer Entropy in Continuous Time Between Neural Spike Trains or Other Event-Based Data

Dear Dr Shorten,

I am pleased to inform you that your manuscript has been formally accepted for publication in PLOS Computational Biology. Your manuscript is now with our production department and you will be notified of the publication date in due course.

With kind regards,

Alice Ellingham
